# Systems biology derived source-sink mechanism of BMP gradient formation

Joseph Zinski[1], Ye Bu[2], Xu Wang[2], Wei Dou[2], David Umulis[2,3]*, Mary C Mullins[1]*

[1]Department of Cell and DevelopmentalBiology, University of Pennsylvania Perelman School of Medicine, Philadelphia, United States; [2]Department of Agricultural and Biological Engineering, Purdue University, West Lafayette, United States; [3]Weldon School of Biomedical Engineering, Purdue University, West Lafayette, United States

**Abstract** A morphogen gradient of Bone Morphogenetic Protein (BMP) signaling patterns the dorsoventral embryonic axis of vertebrates and invertebrates. The prevailing view in vertebrates for BMP gradient formation is through a counter-gradient of BMP antagonists, often along with ligand shuttling to generate peak signaling levels. To delineate the mechanism in zebrafish, we precisely quantified the BMP activity gradient in wild-type and mutant embryos and combined these data with a mathematical model-based computational screen to test hypotheses for gradient formation. Our analysis ruled out a BMP shuttling mechanism and a *bmp* transcriptionally-informed gradient mechanism. Surprisingly, rather than supporting a counter-gradient mechanism, our analyses support a fourth model, a source-sink mechanism, which relies on a restricted BMP antagonist distribution acting as a sink that drives BMP flux dorsally and gradient formation. We measured Bmp2 diffusion and found that it supports the source-sink model, suggesting a new mechanism to shape BMP gradients during development.
DOI: https://doi.org/10.7554/eLife.22199.001

*For correspondence: dumulis@ purdue.edu (DU); mullins@mail. med.upenn.edu (MCM)

**Competing interests:** The authors declare that no competing interests exist.

## Introduction

Morphogen gradients pattern axonal pathways, the neural tube, the dorsal-ventral (DV) and anterior-posterior (AP) embryonic axes, as well as multiple organ systems (*Bökel and Brand, 2013*; *Briscoe and Small, 2015*; *Cohen et al., 2013*; *Rogers and Schier, 2011*; *Rushlow and Shvartsman, 2012*; *Sansom and Livesey, 2009*; *Schilling et al., 2012*; *Tuazon and Mullins, 2015*). Morphogens are defined as factors that form a spatially non-uniform distribution spanning multiple cell-lengths that instructs different cell fates at distinct levels. Their importance in specifying multiple cell fates in a gradient has spurred decades of research deciphering how they work. In 1970, Francis Crick proposed that such a gradient could be formed by a source of morphogen flowing to a sink that destroyed it (*Crick, 1970*). We now know that the mechanisms by which morphogen gradients are established are diverse and complex, and that understanding these mechanisms is paramount to understanding developmental biology (*Briscoe and Small, 2015*; *Müller et al., 2013*; *Rogers and Schier, 2011*). Bone Morphogenetic Proteins (BMPs) act as morphogens repeatedly during development, including in patterning the embryonic DV axis, the neural tube, and the *Drosophila* wing disc (*Bier and De Robertis, 2015*; *Briscoe and Small, 2015*; *Rogers and Schier, 2011*).

BMP morphogen systems are established by a network of extracellular regulators (*Dutko and Mullins, 2011*). A crucial class of these regulators is the BMP antagonists, defined by their ability to bind BMP ligand with high affinity, thereby blocking ligand-receptor interaction (*Brazil et al., 2015*). During axial patterning in zebrafish and *Xenopus*, three antagonists, Chordin, Noggin, and Follistatin play key roles in inhibiting BMP signaling to promote dorsal cell fate specification (*Dal-Pra et al., 2006*; *Khokha et al., 2005*; *Schulte-Merker et al., 1997*). These antagonists bind to the BMP

**eLife digest** Before an animal is born, a protein called BMP plays a key role in establishing the difference between the front and the back of the animal. Cells nearer the front of the embryo contain higher amounts of the BMP protein, whilst cells nearer the back have progressively lower levels of BMP. This gradient of BMP 'concentration' affects the identity of the cells, with the level of BMP in each cell dictating what parts of the body are made where.

The prevailing view among scientists is that the BMP gradient is created by an opposing gradient of another protein called Chordin, which is found at high levels at the back of the embryo and lower levels near the front. Chordin inhibits BMP and the interaction between the two proteins establishes the gradients that create order across the embryo.

Zinski et al. used computer models to investigate how the BMP gradient is created. Several possibilities were considered, including the effect of Chordin. Comparing the models to precise experimental measurements of BMP activity in zebrafish embryos suggested that a different mechanism known as a source-sink model, rather than the opposing Chordin gradient, may be responsible for the pattern of BMP found in the embryo. In this model, the BMP is produced at the front of the embryo and moves towards the back end by diffusion. At the back of the embryo, BMP is mopped up by Chordin, resulting in a constant gradient of BMP along the embryo.

Many other processes that control how animals grow and develop rely on the formation of similar protein gradients, so these findings may also apply to other aspects of animal development. Understanding how animals grow and develop may help researchers to develop strategies to regrow tissues and organs in human patients.

DOI: https://doi.org/10.7554/eLife.22199.002

ligand, preventing BMP from binding its receptors (*Iemura et al., 1998*; *Piccolo et al., 1996*; *Zimmerman et al., 1996*). In both zebrafish and frogs, Chordin differs from Noggin and Follistatin in its expression domain, its phenotype, and its interaction with the metalloprotease Tolloid (Tld). Chordin is expressed in a larger domain than Noggin and Follistatin (*Dal-Pra et al., 2006*; *Khokha et al., 2005*). While the loss of Chordin ventralizes the embryo (*Hammerschmidt et al., 1996*; *Oelgeschläger et al., 2003*), the depletion of Noggin or Follistatin alone or together does not (*Dal-Pra et al., 2006*; *Khokha et al., 2005*). Unlike Noggin and Follistatin, Chordin can be cleaved by the metalloprotease Tolloid, releasing bound BMP ligand and allowing it to signal (*Blader et al., 1997*; *Piccolo et al., 1997*).

Previous studies in *Drosophila* show that the *Drosophila* ortholog of Chordin, Sog, can act as both a BMP agonist and as an antagonist during DV patterning. To act as an agonist, Sog binds to and moves BMP ligand via facilitated diffusion to regions of Tolloid activity (*Figure 1A*). Tolloid then cleaves Sog, which releases BMP thus increasing peak BMP levels, a process altogether known as shuttling (*Figure 1A*) (*Eldar et al., 2002*; *Marqués et al., 1997*; *Holley et al., 1996*; *Peluso et al., 2011*; *Shilo et al., 2013*; *Shimmi et al., 2005*; *Umulis et al., 2010*). The shuttling mechanism is essential to *Drosophila* DV patterning, where Sog shuttles BMP ligand from lateral regions to dorsal regions (*Figure 1A*) (*Eldar et al., 2002*; *Marqués et al., 1997*; *Holley et al., 1996*; *Peluso et al., 2011*; *Shilo et al., 2013*; *Shimmi et al., 2005*; *Umulis et al., 2010*). This shuttling mechanism is required to steepen the BMP signaling gradient and specify the dorsal-most cell fates in the *Drosophila* embryo (*Eldar et al., 2002*; *Marqués et al., 1997*; *Holley et al., 1996*; *Peluso et al., 2011*; *Shilo et al., 2013*; *Shimmi et al., 2005*; *Umulis et al., 2010*). The shuttling of BMP ligand by Chordin has also been suggested to play a role in DV patterning in Echinoderms (*Lapraz et al., 2009*) and Nematostella (*Genikhovich et al., 2015*).

It is unclear whether Chordin shuttles BMP in patterning vertebrate tissues. In *Xenopus*, the shuttling of a particular BMP ligand, ADMP, by Chordin was reported to play a role in DV axial patterning in the scaling of embryos (*Ben-Zvi et al., 2008*; *Reversade and De Robertis, 2005*). In the mouse, Chordin has been suggested to shuttle BMP ligand from where it is expressed in the intervertebral disc to its site of signaling in the vertebral body (*Zakin et al., 2010*). Mathematical models of zebrafish and *Xenopus* DV patterning have predicted that Chordin could shuttle BMP ligand (*Ben-Zvi et al., 2008*; *Zhang et al., 2007*). The transcriptional profiles of zebrafish BMP components at

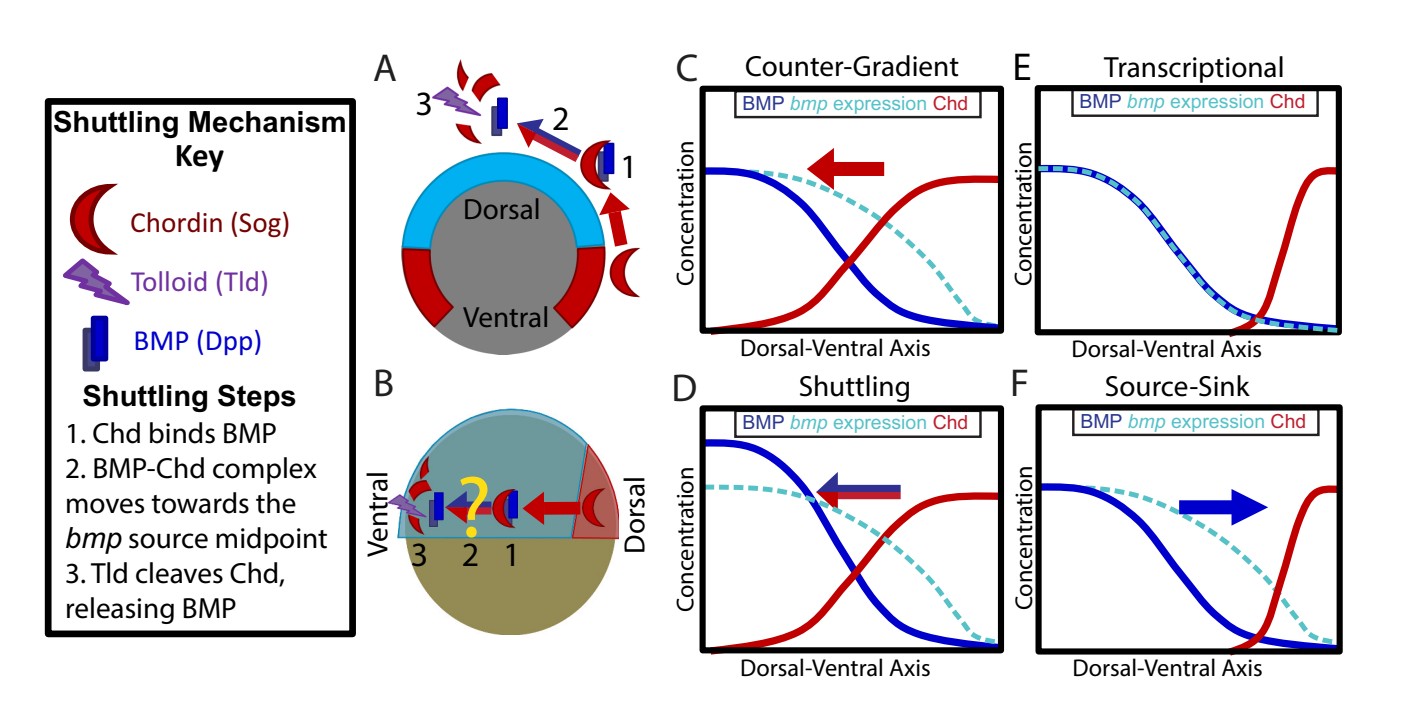

**Figure 1.** Potential Mechanisms of BMP Morphogen Gradient Formation. (**A**) Cross-sectional view of the *Drosophila* embryo depicting Sog shuttling Dpp (the fly BMP ligand) dorsally. (**B**) Lateral view of the zebrafish embryo depicting Chordin (Chd) shuttling BMP ventrally. (**C**) Counter-Gradient: Chd diffuses ventrally to form a counter-gradient repressing BMP. (**D**) Shuttling: BMP bound to Chd is shuttled ventrally, where it is released by Tolloid cleavage. (**E**) Transcriptional: BMP stays where it is produced, mirroring the *bmp* expression gradient. (**F**) Source-sink: BMP diffuses from its source of ventral production to a sink of dorsal Chd.

DOI: https://doi.org/10.7554/eLife.22199.003

the onset of gastrulation resemble that of the *Drosophila* embryo (*Dutko and Mullins, 2011*; *O'Connor et al., 2006*). In *Drosophila*, *sog* is expressed ventral-laterally while the BMP ligand *dpp* is expressed dorsally (*Figure 1A*). Vertebrates have undergone a DV axis inversion with respect to arthropods (*De Robertis and Sasai, 1996*; *Gerhart, 2000*; *Lacalli, 1995*; *Sander and Schmidt-Ott, 2004*), thus *chordin* is expressed dorsally while *bmp* ligands are expressed ventrally (*Figure 1B*). However, whether Chordin acts as a BMP agonist by shuttling BMP ligand during DV patterning in zebrafish or other vertebrates has not been determined (*Figure 1B*).

In vertebrates, the mechanism by which the BMP ligands and antagonists shape this gradient is unclear. Several potential mechanisms have been proposed: 1) an inverse gradient of BMP antagonists imparts the shape of the BMP signaling gradient (*Figure 1C*) (*Blitz et al., 2000*; *Connors et al., 1999*; *Little and Mullins, 2006*; *Thomsen, 1997*), 2) BMP antagonists generate the peak BMP signaling levels by shuttling BMP ligand to these regions (*Figure 1B,D*) (*Ben-Zvi et al., 2008*; *Shilo et al., 2013*; *Zhang et al., 2007*), 3) the gradient shape mirrors the shape of the *bmp* expression domain (*Figure 1E*) (*Ramel and Hill, 2013*), and 4) the gradient is generated by BMP diffusing from its ventral source to a dorsal sink of BMP antagonists (*Figure 1F*). These mechanisms are not mutually exclusive and multiple may act in combination.

To identify the mechanism of BMP signaling gradient formation in the zebrafish embryo, we established a robust quantitative imaging method to directly measure the BMP signaling gradient. We integrated the results with a mathematical modeling approach, using the experiments to inform our model selection. The modeling then provided information on key parameters to measure to identify the mechanism by which the BMP signaling gradient is formed. We used phosphorylated Smad5 protein as a direct read-out for BMP signaling in both wild type (WT), *chordin* mutant, and *chordin* heterozygous embryos. We quantified nuclear phosphorylated-Smad5 (P-Smad5) fluorescent intensity across the entire embryo at single-cell resolution at different stages of development. Combining the P-Smad5 data with a computational screen of mathematical models showed that shuttling

of BMP during DV patterning does not shape the gradient, and that a gradient of *bmp* transcription cannot account for the gradient of BMP signaling activity. From these results, we conclude that the signaling gradient patterning the vertebrate DV axis is generated by either a source-sink or counter-gradient mechanism. To discern between these mechanisms, we developed and measured the diffusion rate of a BMP2-Venus fusion protein in the zebrafish blastula and found that it is relatively mobile, which supports a large number of source-sink simulations, but far fewer counter-gradient simulations. Our results suggest that significant differences exist between the biophysical parameters of conserved proteins in zebrafish and *Drosophila* DV patterning. Through quantification and modeling, we present a new view of the mechanism that the BMP antagonists and ligand use to establish the BMP signaling gradient patterning the DV axis in zebrafish.

## Results

### Quantifying the Wild-Type signaling gradient

To measure the BMP signaling gradient, we quantified the levels of the BMP signal transducer P-Smad5 across the entire embryo at single cell resolution. Smad5 is directly phosphorylated by the BMP type I receptor in response to BMP signaling, and P-Smad5 concentration has been shown to linearly correlate with the concentration of BMP ligand in the *Drosophila* wing disc and S2 cells (*Bollenbach et al., 2008*; *Serpe et al., 2008*). Fixed embryos were whole-mount immunostained for

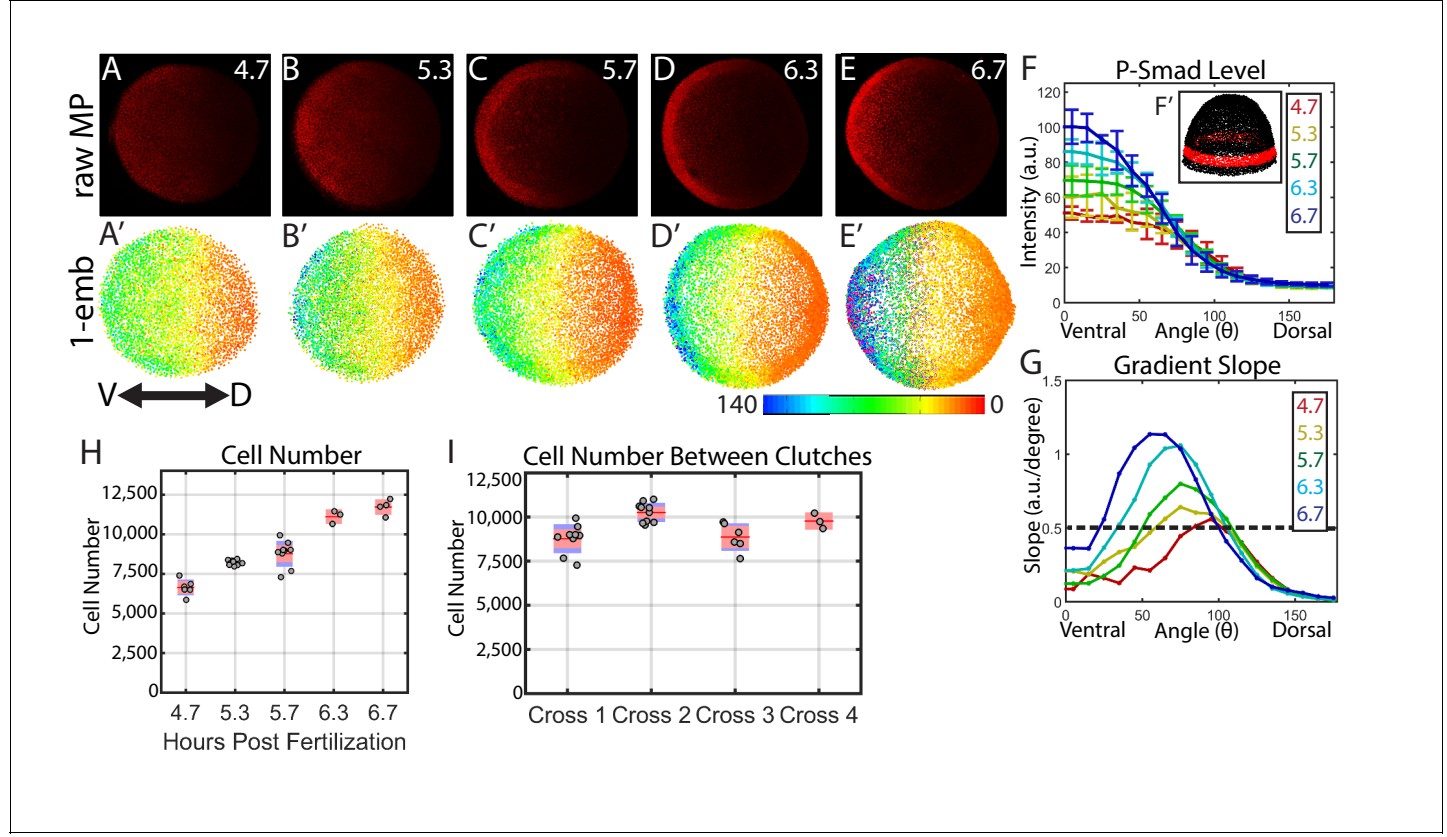

**Figure 2.** Dynamics of the WT P-Smad5 gradient across head and trunk patterning. (**A–E**) Animal views of maximum projections (MP) of P-Smad5 stained individual embryos. (**A'–E'**) Animal views of nuclear intensities of all nuclei from the embryos shown above. (**F**) Average marginal intensities for 4.7–6.7 hpf (4.7: N = 3, 5.3: N = 4, 5.7: N = 13, 6.3: N = 11, 6.7: N = 4). Error bars indicate standard deviation. (**G**) Slope of the P-Smad5 gradients shown in panel F. Dotted line separates high slope (>0.5 a.u./deg) regions from low slope regions. (**H**) Change in cell number versus (vs.) developmental time of embryos fixed from a single cross and nuclei stained with Sytox Orange. (**I**) Cell number varies between different crosses of WT fish fixed at 5.7 hpf. (**H,I**) Gray dots are individual embryo cell counts. Red lines show the mean number of cells at a given time point, red boxes show 95% confidence interval, blue boxes show one standard deviation.
DOI: https://doi.org/10.7554/eLife.22199.004

P-Smad5 and imaged using a Line Scanning Confocal Microscope (*Figure 2A–E*). We developed a mounting and imaging protocol that minimized photo-bleaching, light scattering, and refractive index mismatch (see Materials and methods). We wrote a Matlab algorithm to identify all 8000 + nuclei centerpoints in each embryo in three dimensions, to remove populations unresponsive to P-Smad5 such as yolk syncytial nuclei and dividing cells (see Materials and methods), and to extract the P-Smad5 intensities associated with each nucleus (*Figure 2A'–E'*). Embryos were aligned by coherent point drift to a reference embryo to create ensembles of embryos suitable for statistical analysis (*Myronenko et al., 2010a*) (see Materials and methods). We used a band of cells around the margin of the embryo (*Figure 2F'*) to plot profiles from the dorsal-most to the ventral-most points to compare P-Smad5 gradient profiles between stages and between wild-type and mutant embryos (*Figure 2F*).

Our quantitative analysis revealed that the BMP gradient during DV patterning is quite dynamic. BMP signaling patterns prospective head and rostral trunk DV axial tissues during late blastula to mid-gastrula stages at ~5 to 7 hr post fertilization (hpf) in the zebrafish (*Hashiguchi and Mullins, 2013*; *Kwon et al., 2010*; *Tuazon and Mullins, 2015*; *Tucker et al., 2008*). We quantified the BMP signaling gradient at 30 min intervals across this period. We found that the ventral-most 30° undergoes about a 2-fold intensification from 4.7 to 6.7 hpf (*Figure 2F*). This is accompanied by a 3 to 5 fold increase in the slope of the gradient in ventrolateral regions of the embryo (0–75 degrees) over this 2 hr period (*Figure 2G*). Moreover, the lateral region encompassing the high slope (>0.5 A.U./degree) expands from a size of 20° to 75°, meaning that by 6.7 hpf, nearly half the embryo falls within this high slope region. This contrasts with *Drosophila* DV patterning, where an initial broad, low-slope distribution of P-Mad is refined into a steep peak of BMP signaling covering only the dorsal-most 8% of the embryo (11 cell lengths) (*Sutherland et al., 2003*; *Wang and Ferguson, 2005*). This intensification of P-Mad is very rapid in *Drosophila* DV patterning, where P-Mad increases about three fold in the 30 min (min) between stages 5 and 6 (*Ross et al., 2001*; *Sutherland et al., 2003*; *Wang and Ferguson, 2005*), a process that we found is much slower in the zebrafish embryo: a 2-fold increase over a 2 hr period.

We then sought to determine if changes in cell number could account for the observed changes in the P-Smad5 gradient. We counted the number of cells in each embryo from a single timecourse and observed an approximately 70% increase in cell number from 4.7 to 6.7 hpf (*Figure 2H*). Cell nuclei do not change significantly in size during this time (*Keller et al., 2008*). The increase in cell number occurs throughout the embryo and is not restricted to a particular DV region, while the change in gradient amplitude (*Figure 2F*) and slope is restricted to the ventral third of the embryo (*Figure 2G*). Thus an increase in cell number does not account, via an unknown mechanism, for the increase in amplitude or slope. Additional support that cell number has little effect on gradient shape stems from the observation that the absolute number of cells at a given time point can vary by as much as 20% between different embryos within the same cross or between crosses (*Figure 2I*) with no detectable change in gradient shape or phenotype.

## Mathematical modeling and computational screen of BMP gradient formation

We then performed a computational screen of mathematical models to investigate which gradient-forming mechanisms (*Figure 1C–F*) fit the WT P-Smad5 gradient profiles (*Figure 2*). To do so, we first needed to determine the expression domains of *bmp*, *chordin*, *noggin*, and *tolloid* to use for the mathematical model. We based the domain sizes on our own measurements (*Figure 3*), as well as published in situ hybridizations for *bmp* (*Fürthauer et al., 2004*; *Ramel and Hill, 2013*), *chordin* (*Miller-Bertoglio et al., 1997*), *tolloid* (*Connors et al., 1999*), and *noggin* (*Dal-Pra et al., 2006*). In animal-pole views of wholemount in situ hybridizations, we measured the *chordin* and *noggin* expression domain sizes to be 75 and 40 degrees in width, respectively (*Figure 3A–C*). We estimated the size of the *bmp* expression domain via wholemount in situ hybridizations (*Figure 3D*), however, *bmp2b* expression appeared graded and not as easily measured as *chordin* and *noggin*. We instead measured the *bmp2b* expression gradient at 5.7 hpf via fluorescent in situ hybridization on cross-sections of the DV margin (*Figure 3F–H*). We quantified the relative intensity of the fluorescent *bmp2b in situ* (*Figure 3I* black line) and used it to estimate the BMP production domain in our model (*Figure 3I*, blue line).

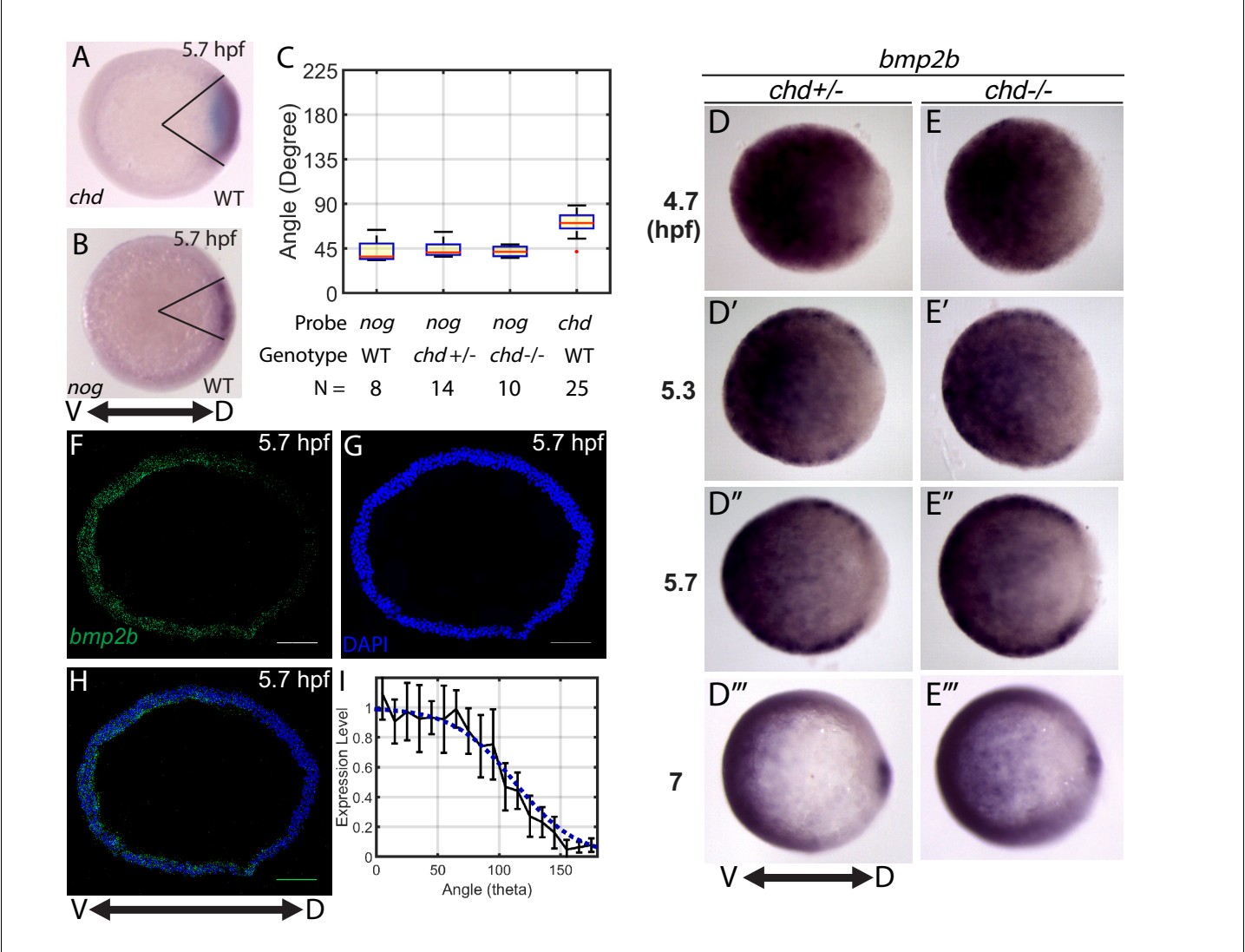

**Figure 3.** Measuring the *bmp2b*, *chordin*, and *noggin* expression domains. Animal pole views of wholemount in situ hybridizations of the expression of (A) *chd* (N = 25), and (B) *nog* (N = 8) in WT embryos. (C) Measured domain size of *chd* and *nog* domains via wholemount in situ hybridization in WT and *chd* mutant embryos. (D–D''') *bmp2b* in *chd* ± embryos at 4.7 (N = 10), 5.3 (N = 15), 5.7 (N = 20), and 7 hpf (N = 16), and (E–E''') *bmp2b* expression in *chd* -/- embryos at 4.7 (N = 6), 5.3 (N = 16), 5.7 (N = 13), and 7 hpf (N = 12). (F–H) Fluorescent in situ hybridization (FISH) signal of *bmp2b* from a marginal slice at 5.7 hpf with a DAPI nuclear stain. Scale bars = 100 µm. (I) Quantification of FISH of *bmp2b* expression from ventral to dorsal (black line, N = 5) compared to the BMP production gradient used in the mathematical model (blue dotted line). Error bars indicate standard deviation.
DOI: https://doi.org/10.7554/eLife.22199.005

We then developed a system of partial differential equations to model the interactions of BMP, Chordin, Noggin, and Tolloid (Supplementary *Figure 4—figure supplement 4 and 1A*). BMP, Chordin, Noggin, BMP-Chordin, and BMP-Noggin were modeled as diffusible species, while Tolloid was treated parametrically according to its domain of expression (*Figure 4A*). The zebrafish gastrula was reduced to a 1-dimensional (1D) half-circumference with a length of 700 µm. We selected a 1D description at the margin for simulation, since the inputs to the system based on gene expression are distributed symmetrically across the embryo, so a 1D model should largely reflect one in 3D, and the faster simulation time of a 1D model allowed us to explore far greater parameter space computationally, increasing our ability to discern biophysical constraints that distinguish between the mechanisms. Domains of production of BMP, Chordin, and Noggin were estimated as described (*Figure 3*, *Figure 4D*). The dissociation constants for BMP-Chordin and BMP-Noggin were set to 1

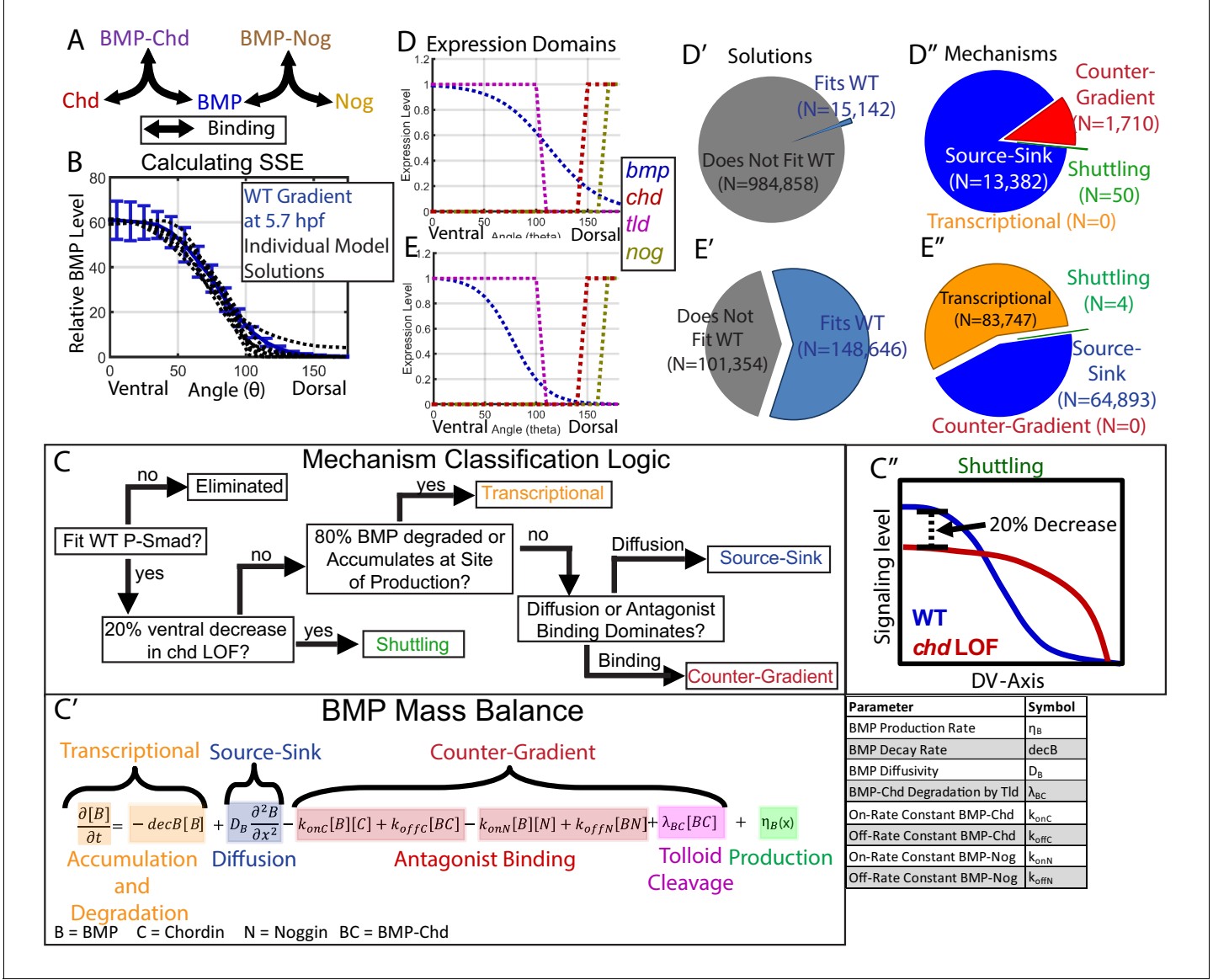

**Figure 4.** Creation of a mathematical model of BMP gradient formation. (A) Depiction of the species and binding possibilities modeled. (B) BMP distributions of 10 individual model solutions (black dotted lines) plotted against the WT 5.7 hpf P-Smad5 gradient (blue line). Error bars indicate standard deviation. (C) Flow-chart of model mechanism classification. (C') BMP mass balance from model labeled to indicate which terms contribute to the source-sink, counter-gradient, and transcriptional mechanisms at each point. (C") Shuttling mechanism was defined by a 20% decrease at the ventral-most point in *chd* LOF compared to WT. (D) Expression domains of *bmp* (blue), *tld* (purple), *chd* (red), and *nog* (yellow) used in the model. (E) Expression domains of *bmp* (blue), *tld* (purple), *chd* (red), and *nog* (yellow) used for the alternative scenario where the *bmp* expression domain mirrors the measured P-Smad5 gradient. (D', E') Pie chart showing how many parameter combinations fit the WT data (blue) and how many failed to do so (grey). (D", E") Pie chart showing how many parameter combinations were classified to have a source-sink (blue), counter-gradient (red), transcriptional (orange), or shuttling (green) mechanism.

DOI: https://doi.org/10.7554/eLife.22199.007

The following figure supplement is available for figure 4:

**Figure supplement 1.** Model-based screen of DV patterning in zebrafish.
DOI: https://doi.org/10.7554/eLife.22199.008

and 0.1 nM respectively, based on previously reported analysis (*Piccolo et al., 1996*; *Troilo et al., 2014*; *Zimmerman et al., 1996*). All remaining parameters (ie. the diffusion coefficients, production rates, decay rates, on and off binding rates) were varied over 4 orders of magnitude encompassing all biologically feasible values (*Table 1*).

**Table 1.** List of the parameter ranges used in the computational model-based screen.

Values range between the upper and lower bound. Note that the dissociation constant of BMP-Chd and BMP-Nog was held constant, but the on- and off- rates were allowed to vary.

| Parameter | Units | Symbol | Lower bound | Upper bound |
|---|---|---|---|---|
| BMP Production Rate | nM/s | $\eta_B$ | $10^{-2}$ | $10^{2}$ |
| BMP Decay Rate | 1/s | decB | $10^{-1}$ | $10^{-5}$ |
| BMP Diffusivity | $\mu m^2/s$ | $D_B$ | $10^{-2}$ | $10^{2}$ |
| Chd Production Rate | nM/s | $\eta_C$ | $10^{-2}$ | $10^{2}$ |
| Chd Decay Rate | 1/s | decC | $10^{-1}$ | $10^{-5}$ |
| Chd Diffusivity | $\mu m^2/s$ | $D_C$ | $10^{-2}$ | $10^{2}$ |
| Nog Production Rate | nM/s | $\eta_N$ | $10^{-2}$ | $10^{2}$ |
| Nog Decay Rate | 1/s | decN | $10^{-1}$ | $10^{-5}$ |
| Nog Diffusivity | $\mu m^2/s$ | $D_N$ | $10^{-2}$ | $10^{2}$ |
| BMP-Nog Decay Rate | 1/s | decBN | $10^{-1}$ | $10^{-5}$ |
| BMP-Nog Diffusivity | $\mu m^2/s$ | $D_{BN}$ | $10^{-2}$ | $10^{2}$ |
| BMP-Chd Decay Rate | 1/s | decBC | $10^{-1}$ | $10^{-5}$ |
| BMP-Chd Diffusivity | $\mu m^2/s$ | $D_{BC}$ | $10^{-2}$ | $10^{2}$ |
| Chd Degradation by Tld | 1/s | $\lambda_C$ | $10^{0}$ | $10^{-4}$ |
| BMP-Chd Degradation by Tld | 1/s | $\lambda_{BC}$ | $10^{0}$ | $10^{-4}$ |
| Length of the Embryo | $\mu m$ | - | 700 | 700 |
| Length of the Chd domain (from dorsal) | $\mu m$ | - | 145 | 145 |
| Length of the Nog domain (from dorsal) | $\mu m$ | - | 78 | 78 |
| Length of the Tolloid domain (from ventral) | $\mu m$ | - | 400 | 400 |
| Dissociation Constant of BMP-Chd | nM | - | 1 | 1 |
| Dissociation Constant of BMP-Nog | nM | - | 0.1 | 0.1 |
| Time | min | t | 130 | 130 |

DOI: https://doi.org/10.7554/eLife.22199.006

The equations were solved for the developmental window from ~3.5 hpf to ~5.7 hpf, since *bmp* and *chordin* are first expressed shortly after the mid-blastula-transition at 3 hpf (*Koos and Ho, 1999*; *Leung et al., 2003*; *Shimizu et al., 2000*; *Solnica-Krezel et al., 2001*). The equations were simulated 1,000,000 times, each time with a different combination of randomly selected parameters. Each parameter combination was then re-simulated without Chordin or Noggin to predict the BMP signaling gradient in a *chordin* or *noggin* loss-of-function (LOF) scenario. We then selected simulations that generated BMP profiles that fit our measured P-Smad5 gradient at 5.7 hpf with a normalized root mean squared deviation of 8% or less (*Figure 4B*). We also eliminated simulations that had significantly different BMP profiles when Noggin production was set to 0, since the loss of Noggin does not affect DV patterning in zebrafish or *Xenopus* (*Dal-Pra et al., 2006*; *Khokha et al., 2005*).

All simulation results that fit our data were classified into categories based on the biophysical process that dominated formation of the gradient shape: shuttling, source-sink, counter-gradient, or transcriptional (*Figure 4C*). We discerned between source-sink, counter-gradient, and transcriptional mechanisms by examining the balance of binding, diffusion, decay, and accumulation processes in the partial differential equation for the BMP species (*Figure 4C–C''*). If 80% of the BMP ligand was degraded where it was produced or accumulated there, the simulation was classified as transcriptional (*Figure 4C,C'*). If the majority of BMP diffused away from its site of production rather than being bound by Chordin, the simulation was considered a source-sink mechanism (*Figure 4C,C'*). Conversely, if the majority of BMP was bound at its site of production by Chordin, the simulation was classified as a Chordin counter-gradient mechanism (*Figure 4C,C'*). We classified a simulation as shuttling if the ventral-most point in the predicted *chordin-/-* BMP profile was at least 20% lower than in WT, as shuttling by the antagonist leads to a net accumulation of ligand in the ventral-most

region (*Figure 4C,C''*). By comparison, shuttling in *Drosophila* accounts more significantly to the peak level, with a 50% decrease in the peak P-Mad level when the *chordin* homolog *sog* is deficient (*Mizutani et al., 2005*; *Peluso et al., 2011*; *Sutherland et al., 2003*). Shuttling of 20% is about one standard deviation greater than our measured embryo-to-embryo variability for peak P-Smad5 signaling levels (*Figure 4B*).

Multiple classes of mechanisms generated simulations fitting our WT data, including an antagonist counter-gradient, source-sink, and shuttling. Of the 1,000,000 randomly picked parameter combinations, 15,142 fit the experimentally measured WT signaling gradient (*Figure 4D'*). Among those that fit, 13,382 were classified as source-sink, 1710 as counter-gradient, and 50 as shuttling (*Figure 4D''*). Notably, no transcriptional simulations were found, because our measured *bmp* expression profile (*Figure 3I*) did not match our measured WT BMP signaling gradient (*Figure 2*), and therefore BMP must diffuse from its site of production or be bound by Chordin to fit our measured signaling gradient.

The mean of our *bmp2b* expression profile did not reflect the P-Smad5 gradient at 5.7 hpf (*Figures 2F* and *3I*), however, the *bmp2b* expression profile displayed variability from embryo to embryo. To account for the possibility that the *bmp* expression profile reflects the WT P-Smad5 gradient, which fell within one standard deviation of the mean (*Figures 2F* and *3I*), we simulated the mathematical model 250,000 additional times with a *bmp* expression input matching the WT P-Smad5 gradient (*Figure 4E*). We found that the transcriptional mechanism was the most abundant mechanism among the simulations, comprising 83,747 of the simulations (*Figure 4E',E''*). Surprisingly, no counter-gradient solutions were found when *bmp* expression matched P-Smad5. This is because Chordin binding to BMP would interfere with the shape of the BMP protein gradient, causing it to no longer match the measured BMP signaling gradient. Thus the transcriptional and counter-gradient mechanisms are incompatible with each other.

## Constraints on biophysical parameters imposed by the WT gradient

Each mechanism required specific biophysical parameters to fit the experimentally measured DV signaling gradient. The source-sink mechanism required BMP to have a high diffusion rate, so BMP could diffuse to a dorsally-localized sink of antagonists (*Figure 5A*). The counter-gradient mechanism required Chordin to have a high diffusion rate, so Chordin could diffuse ventrally to generate an antagonist gradient (*Figure 5B*). When either BMP or Chordin had moderately high diffusion rates (e.g. near 1 um$^2$/s), decay rates were low to allow each more time to diffuse dorsally or ventrally, respectively. When BMP and Chordin range are plotted on the same axis, the segregation of the source-sink and counter-gradient mechanisms based on range is readily apparent (*Figure 5C*). The shuttling mechanism required that BMP-Chordin have a high diffusion rate and low decay rate in order to diffuse ventrally where Tolloid cleaves Chordin, which then releases BMP (*Figure 5D*). Noggin diffusion and decay was not constrained in any of the three mechanisms (*Figure 5E*).

For the simulations performed where the *bmp* expression domain matched the signaling gradient, distinct biophysical requirements were also observed for each mechanism. The source-sink mechanism required BMP to have a high range, while the transcriptional mechanism required BMP to have a lower range (*Figure 5I*). The source-sink mechanism required either high diffusivity with predominantly high decay rates or low diffusivity with low decay rates, while the transcriptional mechanism filled a near complementary parameter space with progressively higher BMP diffusivity necessitating higher decay rates (*Figure 5G*), which allows the P-Smad5 gradient to match the *bmp* expression profile. Neither of these mechanisms constrained Chordin range (*Figure 5H,I*). Although only four shuttling mechanism solutions were identified, they required both BMP-Chordin range to be high and its decay rate to be low (*Figure 5J*). The shuttling solutions also required BMP range to be high (*Figure 5G,I*), because if Chordin moved ventrally to bind BMP in this scenario, it would interfere with the *bmp* expression gradient matching the BMP signaling gradient. In these simulations BMP moves dorsally to form the BMP-Chordin species that ultimately is shuttled back ventrally. The remaining parameters required for shuttling were similar for the two simulations (*Figure 5E,F,K, L*).

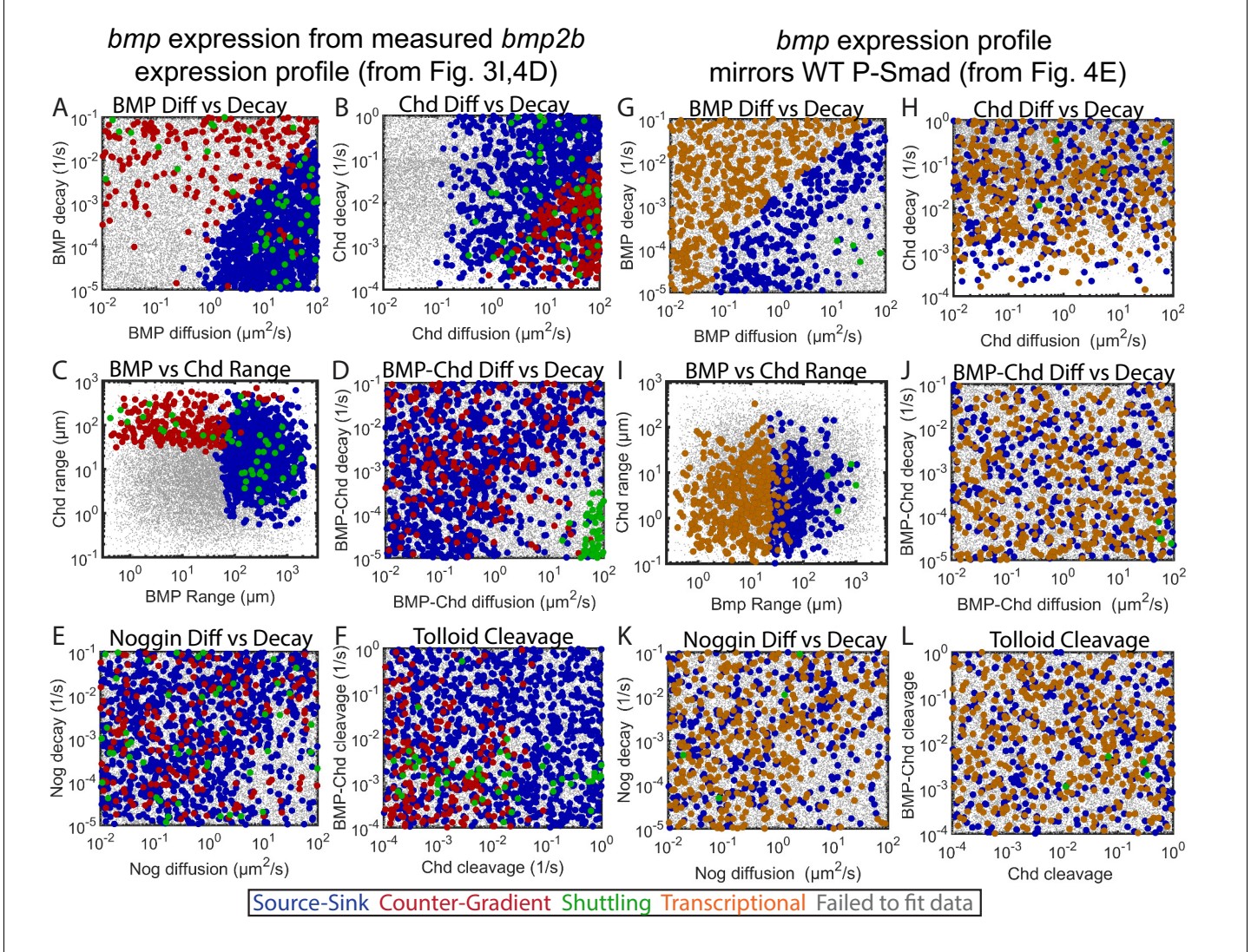

**Figure 5.** Biophysical values of individual simulations that fit the WT P-Smad5 gradient. (A–L) Scatter plots comparing biophysical parameters of 1000 randomly selected solutions classified by mechanism that fit the WT data. Combinations that failed to fit the WT P-Smad5 gradient are small grey dots. We plot solutions as large circles colored according to their mechanism, which is based on definitions outlined in *Figure 4C*: counter-gradient (red), source-sink (blue), transcriptional (orange), or shuttling (green). We plotted additional shuttling solutions in order to better illustrate trends. (A–F) Simulations using domains displayed in *Figure 4D*. (G–L) Simulations using domains displayed in *Figure 4E*. (A,G) BMP diffusivity vs. BMP decay rate. (B,H) Chd diffusivity vs. Chd decay rate, which includesthe rate of Chd cleavage by Tld. (C,I) Range was estimated as sqrt(diffusivity/decay). (D,J) Diffusivity of BMP bound to Chd vs. decay rate of BMP bound to Chd. (E,K) Range of Nog protein. (F,L) Chd and BMP-Chd cleavage rate by Tld.
DOI: https://doi.org/10.7554/eLife.22199.009

## The *chordin* mutant gradient shows no evidence of shuttling

To determine whether Chordin shuttling of BMP ligand plays a functionally relevant role in generating the ventral P-Smad5 peak in zebrafish, as it does in *Drosophila*, we quantified the P-Smad5 gradient of *chordin* mutant embryos over a developmental time series (*Figure 6A,B*). If it does play a role, then we expect the ventral P-Smad5 peak to be reduced in *chordin* mutants compared to WT embryos. We found that the P-Smad5 gradient in *chordin* mutants showed a statistically significant increase in lateral regions of the embryo at the four time-points examined from 4.7 to 6.3 hpf (*Figure 6C–F*, *Figure 6—figure supplement 1A*). Importantly, no decrease in P-Smad5 was observed in the ventral region of *chordin* mutant embryos or in any region of the gradient. These results indicate that, unlike the *Drosophila* homolog Sog, Chordin plays no significant BMP shuttling

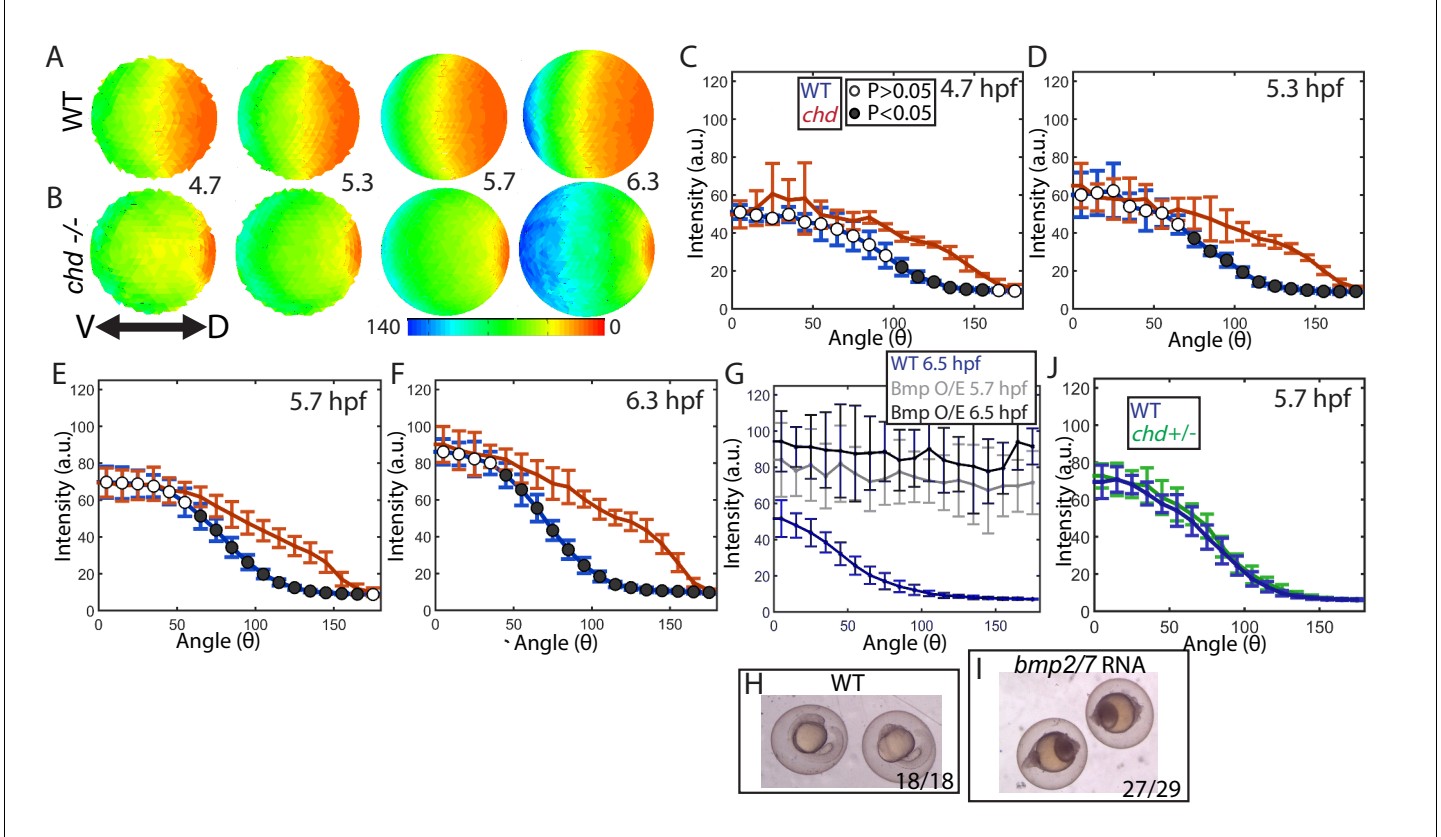

**Figure 6.** Effect of Chd on gradient shape and ligand shuttling. (**A,B**) Animal views of average intensities from each time-point in (**A**) WT (4.7: N = 3, 5.3: N = 4, 5.7: N = 13, 6.3: N = 11) and (**B**) *chd* mutant (4.7: N = 3, 5.3: N = 5, 5.7: N = 11, 6.3: N = 9) embryos. (**C–F**) Average marginal intensities for WT (blue) and *chd* mutant (red) embryos from 5.7 to 6.3 hpf. (**G**) Average marginal intensities for WT (blue, N = 4) and *bmp2/7* RNA injected embryos at 5.7 (grey, N = 4) and 6.3 hpf (black, N = 5). Error bars indicate standard deviation. (**H,I**) Fully ventralized (V5) embryos injected with 6 pg of *bmp7a* RNA and 12 pg of *bmp2b* RNA vs uninjected WT siblings. (**J**) WT (N = 9) vs *chd+/-* (N = 10) at 5.7 hpf.

DOI: https://doi.org/10.7554/eLife.22199.010

The following figure supplement is available for figure 6:

**Figure supplement 1.** Effect of Chd on gradient shape and ligand shuttling.

DOI: https://doi.org/10.7554/eLife.22199.011

role during zebrafish DV patterning. It is worth noting that in many simulations, small amounts of BMP ligand are shuttled short distances but do not impact the gradient significantly, and thus are not classified as shuttling. The P-Smad5 gradient in *chordin* mutants shows that this is minimal in zebrafish.

Interestingly, the loss of *chordin* did not cause an increase in the ventral-most P-Smad5 peak level either (**Figure 6C–F**,S1A), suggesting that Chordin does not actively block BMP signaling there. However, Smad5 or another signal transducing component could be limiting in the ventral-most cells of WT embryos, rendering them unresponsive to further increases in free ligand. To investigate this possibility, we overexpressed Bmp2/7 ligand in WT embryos at a level that fully ventralizes (V5) them at 24 hpf (**Figure 6H,I**). We quantified P-Smad5 at 5.7 and 6.5 hpf and found that the gradient showed a significant increase in signaling embryo-wide over WT siblings, including in the ventral-most region (**Figure 6G**). These results indicate that BMP signaling in ventral regions is not saturated in WT embryos and that Chordin does not regulate the peak P-Smad5 levels by promoting or inhibiting signaling at these stages.

We next tested whether the P-Smad5 gradient is robust to heterozygosity of *chordin*. The *Drosophila* P-Mad gradient shows some small changes in *sog* heterozygotes (*sog* is the *chordin* homolog) compared to WT (**Eldar et al., 2002**; **Umulis et al., 2010**). In zebrafish *chordin* heterozygotes

do not show any DV patterning phenotype at 24 hpf (*Hammerschmidt et al., 1996*). Consistent with this, we found that the *chordin* heterozygous and WT BMP signaling gradients were indistinguishable at 5.7 hpf (*Figure 6J*). Therefore, the BMP signaling gradient in zebrafish is robust to *chordin* heterozygosity and to a potential 50% decrease in Chordin levels, but not to the complete loss of Chordin.

## Constraints on computational models imposed by the *chordin* mutant gradient

We then constrained the mathematical model-based computational screen with the WT, the *chordin* mutant and heterozygote P-Smad5 gradients at 5.7 hpf, assuming that heterozygotes have half the Chordin level of WT. We determined which mechanisms and simulations remained compatible with these new results (*Figure 7A,B*). We eliminated simulations that deviated by more than 8% from the *chordin* heterozygous or homozygous P-Smad5 gradients (*Figure 7C–D*). Many mathematical model simulations that fit the WT BMP signaling gradient no longer fit our *chordin* heterozygous and homozygous mutant data. Of the 15,142 simulations that fit the WT gradient alone, only 4059 fit the WT, *chordin +/-*, and *chordin* mutant gradients (*Figure 7E–E'*). Of those, all were either source-sink or counter-gradient mechanisms (*Figure 7E'*). All simulations classified as shuttling had *chordin* LOF BMP distributions that deviated from the measured *chordin* mutant P-Smad5 gradient by more than 17% (*Figure 7C*). Many, but not all, simulations classified as counter-gradient deviated by more than 8% in their BMP distributions from the measured *chordin* heterozygous P-Smad5 gradient (*Figure 7D*). Fitting the *chd* LOF and *chd* heterozygous data eliminated more counter-gradient than source-sink solutions, with 28% of the source-sink solutions remaining but only 15% of the counter-gradient solutions remaining.

The remaining mechanisms required different and specific combinations of biophysical parameters. The source-sink solutions required a high BMP range of 60+ μm with a diffusivity above 1 μm$^2$/s (*Figure 7F,G*). The counter-gradient mechanism required a lower BMP range, less than 60 μm, a high Chordin range above 40 μm with a diffusivity above 2 μm$^2$/s (*Figure 7G,H*). Consistent with this, very high rates of Chordin cleavage by Tolloid restricted Chordin range and therefore was not compatible with the counter-gradient mechanism (*Figure 7I*). While BMP diffusivity needed to be high in source-sink solutions, BMP-Chd diffusivity was not restricted in either the source-sink or counter-gradient solutions (*Figure 7J*).

We then tested whether the transcriptional mechanism could also fit the *chordin* mutant data. We used the simulation in which the *bmp* expression profile matched the WT P-Smad5 gradient (*Figure 4E*). Of the 148,646 simulations that fit the WT data alone, 227 fit the WT, *chordin* heterozygous, and *chordin* mutant gradients, all of which were source-sink (*Figure 7K,K'*). These source-sink simulations required a high BMP diffusivity and range (*Figure 7L,M*), similar to what was observed in the simulation with the measured *bmp* expression profile (*Figure 7F,G*). However, while 83,747 simulations fit a transcriptional mechanism with the WT data alone, none fit the WT, *chordin* heterozygous, and *chordin* mutant gradients (*Figure 7K,K'*). This is because only a change in the *bmp* expression domain in the *chordin* mutant allows a transcriptional mechanism to fit both the WT and *chordin* mutant data.

The *bmp* expression domain is known to become responsive to BMP signaling, creating a positive feedback loop during gastrulation (*Hammerschmidt et al., 1996*; *Nguyen et al., 1998*; *Schmid et al., 2000*). Gastrulation begins in zebrafish at 6 hpf. While the initial *bmp* expression domains are established independently of BMP feedback, a BMP feedback loop becomes active with reported onset times ranging from ~5.5 to 6.5 hpf (*Kishimoto et al., 1997*; *Miller-Bertoglio et al., 1999*; *Ramel and Hill, 2013*; *Schmid et al., 2000*). To test whether the *bmp* expression domain changes in *chordin* mutants at 4.7 to 5.7 hpf, we compared the *bmp2b* domain size in sibling *chordin-/-* and *chordin +/-* embryos (*Figure 3D,E*). *chordin* heterozygotes display a WT phenotype (*Hammerschmidt et al., 1996*; *Miller-Bertoglio et al., 1999*) and we found that they also display a WT P-Smad5 gradient (*Figure 6J*). There was no discernable difference in the *bmp2b* domain size at 4.7, 5.3, or 5.7 hpf (*Figure 3D,E*), indicating that *bmp* transcriptional feedback is not active before 5.7 hpf. Similarly, the *noggin* expression domain size did not change in *chordin* mutants before 5.7 hpf (*Figure 3C*). Therefore, the increase in BMP signaling activity observed already at 4.7 hpf in *chordin* mutants (*Figure 6A–C*) precedes a change in *bmp* expression, showing that the transcriptional mechanism cannot account for the P-Smad5 gradient profiles prior to 5.7

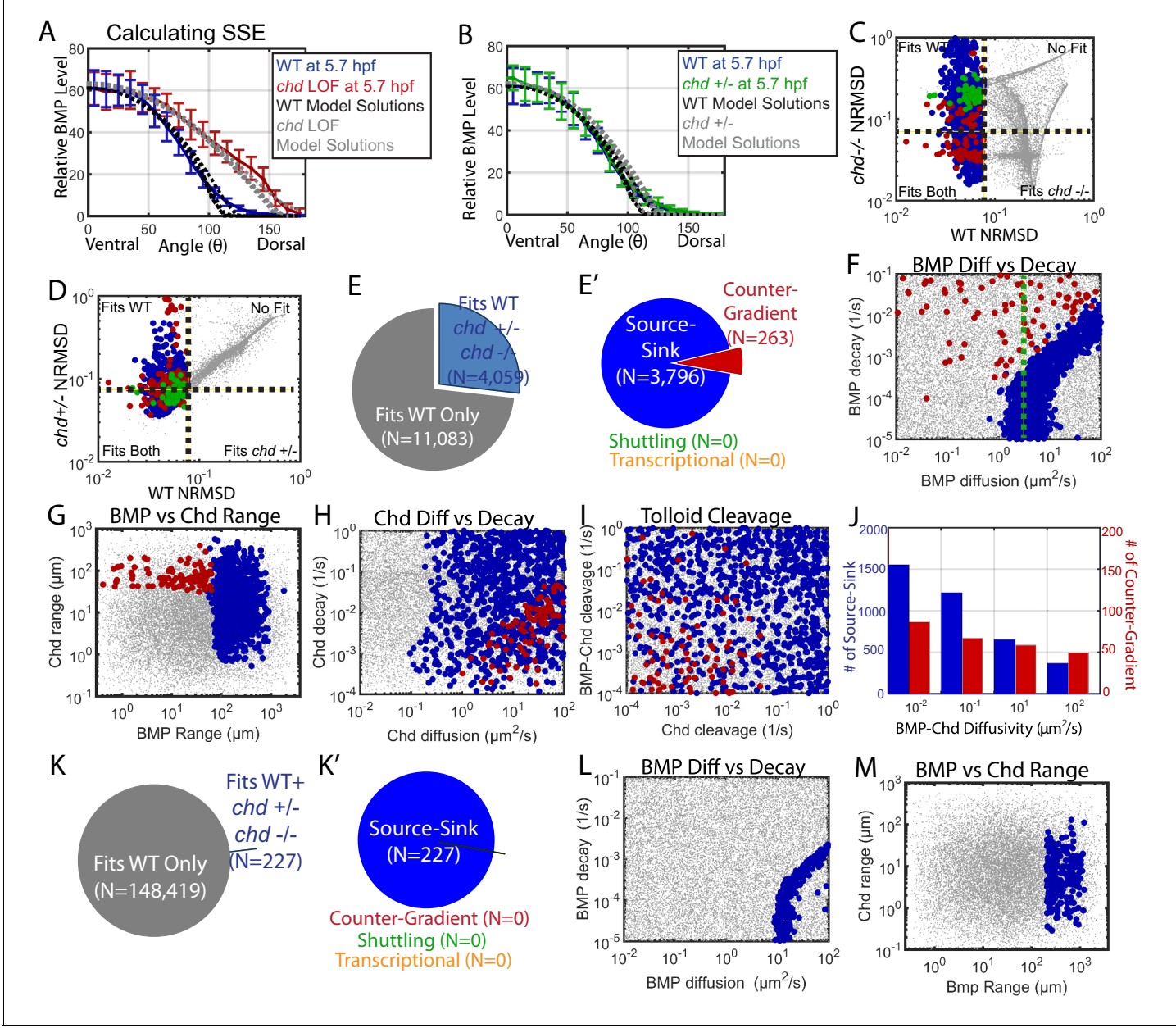

**Figure 7.** Biophysical values of individual simulations that fit both the WT and *chd* LOF P-Smad5 gradients. (A) BMP distributions of 5 individual modeling solutions (WT: black dotted lines, *chd* LOF: grey dotted lines) plotted against WT (blue line) and *chd* LOF (red line) 5.7 hpf P-Smad5 gradients. Error bars indicate standard deviation of experimental P-Smad5 intensity. (B) BMP distributions of 5 individual modeling solutions (WT: black dotted lines, *chd* +/-: grey dotted lines) plotted against WT (blue line) and *chd* +/- (green line) 5.7 hpf P-Smad5 gradients. Error bars indicate standard deviation of experimental P-Smad5 intensity. (C,D,F–J,L, M) 1000 randomly selected parameter combinations capable of fitting both the WT data, *chd* +/-, *chd* LOF data classified by mechanism. Larger circular points fit the WT P-Smad gradient and are colored based on their mechanism according to the definitions outlined in *Figure 4C*: counter-gradient (red), source-sink (blue), transcriptional (orange), or shuttling (green). Combinations that failed to fit the WT P-Smad5 gradient are small grey dots. (C–D) Normalized Root Mean Squared Deviation (NRMSD) between the measured P-Smad5 and the model BMP distributions. Black dotted lines mark the 8% threshold. (C) Comparing WT and *chd* LOF. (D) Comparing WT and *chd* +/-. (E) Parameter combinations that fit both the WT and *chd* LOF data (blue) and those that failed to do so (grey). (E') parameter combinations were classified to have a source-sink (blue), counter-gradient (red), or shuttling (green) mechanism. (F–J) Simulation using the *bmp* expression domain displayed in *Figure 4D*. (L,M) Simulation using the *bmp* expression domain displayed in *Figure 4E*. (F,L) BMP diffusivity vs. BMP decay rate. Green dotted line marks the BMP diffusivity we measured using FRAP (4.4 μm²/s). (G,M) Range was estimated as sqrt(diffusivity/decay). (H) Chd diffusivity vs. Chd decay rate plus the rate of Chd cleavage by Tld. (I) Rate of Chd cleavage by Tld vs rate of BMP-Chd cleavage by Tld. (J) BMP-Chd diffusivity for source-sink and counter-gradient simulation solutions. (K) Pie chart showing the parameter combinations that fit the WT data (blue) or failed to do so (grey) for the alternative

*Figure 7 continued on next page*

*Figure 7 continued*

scenario where the *bmp* expression domain mirrors the measured P-Smad5 gradient (*Figure 3I*). (K') Pie chart of the solutions that had a source-sink (blue), counter-gradient (red), transcriptional (orange), or shuttling (green) mechanism.

DOI: https://doi.org/10.7554/eLife.22199.012

hpf. We observed a change in *bmp2b* expression by 7 hpf, consistent with previous findings that *bmp* transcriptional feedback activates after gastrulation begins (*Figure 3D''',E'''*) (*Kishimoto et al., 1997*; *Miller-Bertoglio et al., 1999*; *Ramel and Hill, 2013*; *Schmid et al., 2000*).

## Fluorescence recovery after photobleaching to measure BMP diffusivity

The combination of WT and *chordin* mutant P-Smad5 gradients limits the number of computationally-derived model simulations and, importantly, reduces the number of mechanisms to two: source-sink and counter-gradient. We and others have largely purported the counter-gradient mechanism as acting in vertebrate DV patterning (*Blitz et al., 2000*; *Hama and Weinstein, 2001*; *Little and Mullins, 2006*; *Sasai and De Robertis, 1997*; *Thomsen, 1997*). To our surprise, the source-sink modeling simulations emerged more frequently within our computational screen than the counter-gradient simulations (*Figure 7E'*). To test the source-sink mechanism further, we investigated BMP ligand diffusivity, a biophysical parameter that must be high in this mechanism (*Figure 7F*). To test if BMP diffusivity excludes or supports the source-sink mechanism, we measured the effective diffusivity of the Bmp2b ligand using fluorescence recovery after photobleaching (FRAP).

We tagged Bmp2b by inserting the coding sequence of the fluorescent protein Venus between the pro- and mature coding domains of *bmp2b*. When mRNA of *bmp2b-venus* was injected into 1-cell stage zebrafish embryos, we detected both the pro- and mature domains of Bmp2b-Venus protein on a western blot (*Figure 8A*, black arrows, n = 3). We did not detect protein with a molecular weight equal to Venus alone in the embryos injected with *bmp2b-venus*, indicating that the Venus tag was not cleaved from Bmp2b during post-translational processing (*Figure 8A*, red arrow). The Venus tag did not interfere with Bmp2b activity, since the injected mRNA significantly ventralized WT embryos (*Figure 8B*, Row 1). To further assess the activity and range of the Bmp2b-Venus chimera, we tested if Bmp2b-Venus could rescue embryos deficient in Bmp2b. As previously reported for *bmp2b* RNA (*Nguyen et al., 1998*), we could rescue Bmp2b deficient embryos to a WT phenotype by injecting *bmp2b-venus* RNA (*Figure 8B*, Rows 5–7).

To perform the FRAP, we injected *bmp2b-venus* mRNA into a single blastomere at the 8 cell stage (*Figure 8B*, Row 3) and then photobleached a 160 μm cube of cells in 4.3 hpf embryos (*Figure 8C*). We then measured recovery of fluorescence over one hour. To ensure we only recorded extracellular Bmp2b-Venus, we photobleached a region away from the cells producing Bmp2b-Venus. The bleached region recovered fluorescence to its initial level in ~30 min (*Figure 8C,E*), corresponding to a measured Bmp2b-Venus effective diffusivity of 4.4 ± 0.4 μm$^2$/s (SEM (standard error of the mean), n = 5). To ensure that we were only measuring the diffusivity of Bmp2b-Venus alone and not Bmp2b-Venus bound to Chordin, we repeated the FRAP experiment in Chordin deficient embryos (*Figure 8B*, Rows 3–4). Again, the bleached region recovered fluorescence to its initial level in ~30 min (*Figure 8F*), corresponding to a measured Bmp2b-Venus effective diffusivity of 4.0 ± 0.5 μm$^2$/s (SEM, n = 5). To determine the extent to which Venus limited Bmp2b diffusion, we measured the diffusivity of Venus alone. The bleached region recovered fluorescence much more rapidly, and neared its initial level in 5 min (*Figure 8D,G*), corresponding to a measured Venus effective diffusivity of 16.3 ± 2.2 μm$^2$/s (SEM, n = 5). This faster rate of Venus diffusion indicates that it per se does not reduce BMP2 diffusion.

A measured BMP diffusivity of ~4 μm$^2$/s fits a large portion of the source-sink modeling simulations. In fact, 1421 source-sink simulations have diffusivities within 2 μm$^2$/s of our measured diffusivity (*Figure 8H*). In contrast, only 31 counter-gradient simulations were within 2 μm$^2$/s of our measured diffusivity, and these simulations have very high BMP decay rates (above 10$^{-3}$/s) (*Figure 8H*). A decay rate of that magnitude would cause the half-life of BMP ligand in the embryo to be very short, <10 min for decay rates above $1 \times 10^{-3}$/s. We can infer the BMP lifetime in the embryo based on our previous studies. We found previously that a pulse of injected BMP ligand protein in a *bmp* deficient embryo sustains P-Smad5 for at least 1.5 hr after injection (*Little and Mullins,*

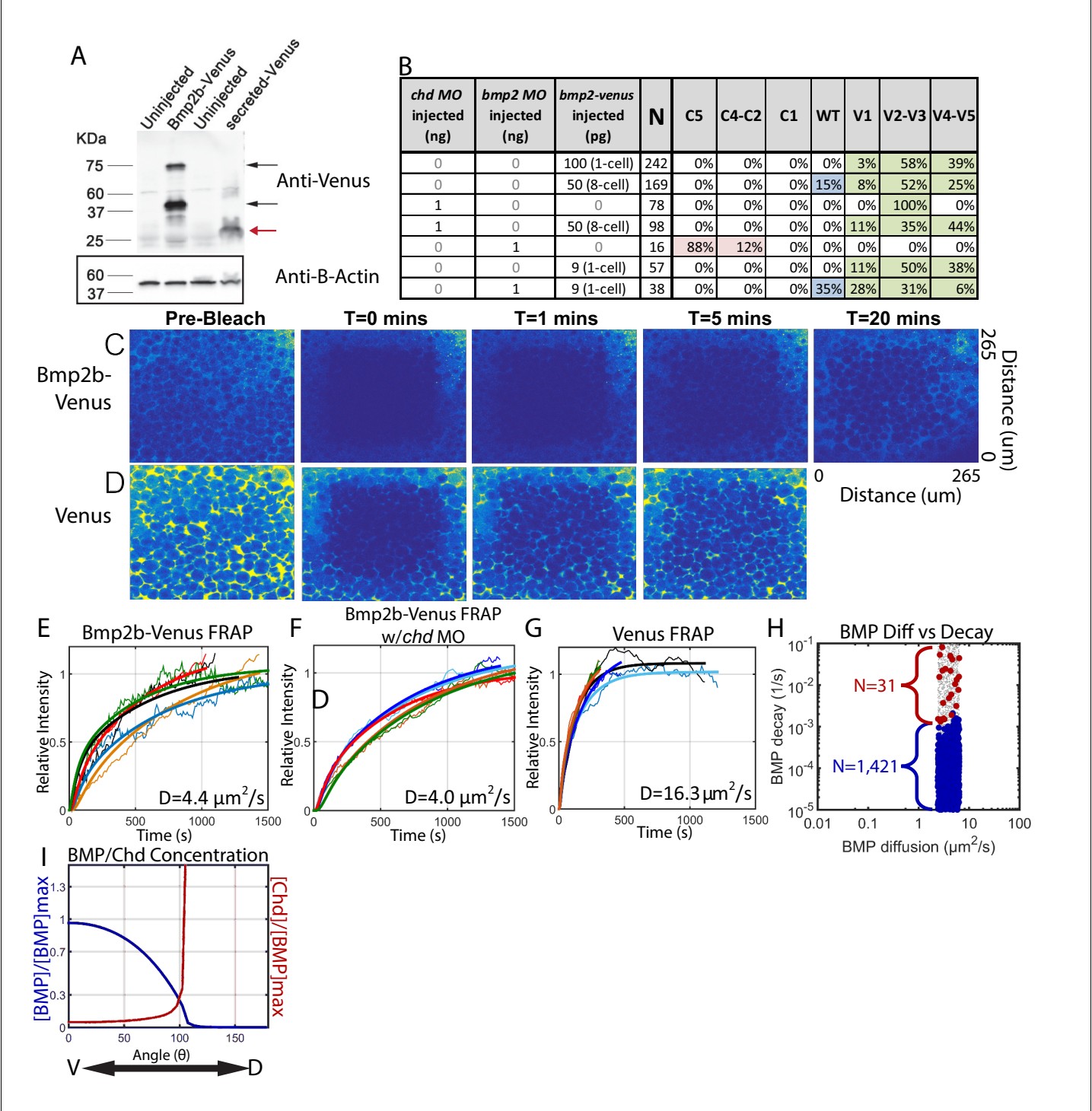

**Figure 8.** Measuring Bmp2b-Venus diffusivity via FRAP. (**A**) Detection of Bmp2b-Venus and secreted Venus proteins by western blot. Embryos were injected with *bmp2b-venus* mRNA (250 pg) or secreted-Venus mRNA (200 pg) at the one-cell stage. Protein lysates were prepared at late blastula stage. In the Bmp2b-Venus overexpression sample, two major protein bands were detected by Venus antibody (black arrows). The larger molecular weight protein is the pro- and mature domains of Bmp2b with Venus protein (669 amino acids (AA),~74 KDa). The smaller protein is the mature domain of Bmp2b with Venus protein (376 AA,~41 KDa). The secreted Venus protein (248 AA,~27 KDa) is also detected in the secreted-Venus overexpression sample (red arrow). β-actin was used as a loading control. (**B**) 24 hpf phenotypes of embryos injected with the *bmp2b-venus* construct used for FRAP experiments, controls, and rescue. Dorsalization was classified as C5: Loss of all ventral structures; C4-C3: Loss of, or truncated tail; C2-C1: Loss of ventral tail fin. Ventralization is classified as V1: reduction is eye size; V2-V3: the eyes, notochord, and anterior brain are partially or completely absent; or V4-V5: complete loss of all dorsal structures. Fluorescent BMP-Venus (**C**) or Venus (**D**) recovery after photobleaching for 20 min. (**E–G**) Plots of

*Figure 8 continued on next page*

*Figure 8 continued*

fluorescent intensity recovery in the extracellular region. Bold lines are mean curves, thin lines are raw intensity data. (H) BMP diffusivity vs. BMP decay rate for simulations that fit WT, *chd +/-*, and *chd -/-* P-Smad5 profiles and were within 2 μm²/s of 4.4 μm²/s. Large blue circles are simulations classified as source-sink, red are counter-gradient, and small grey dots failed to fit the measured P-Smad5 profiles. (I) The mean BMP and Chd concentrations in all solutions that fit the WT, *chd-/-*, and *chd +/-* P-Smad5 data and within a diffusivity of 2.4 and 6.4 μm²/s that are also robust to uniform Chd production.

DOI: https://doi.org/10.7554/eLife.22199.013

*2009*). In another study, we found that P-Smad5 can persist for a maximum of 40–60 min in the absence of BMP signaling (*Tucker et al., 2008*). Thus, P-Smad5 persistence 1.5 hr after a BMP protein pulse indicates that the ligand remains for at least 30 to 50 min after injection, if P-Smad5 can persist for 40–60 min in the absence of signaling. A BMP ligand lifetime of 30 to 50 min is inconsistent with the low BMP half-lives required for the counter-gradient simulations, but consistent with the BMP half-lives of source-sink simulations (*Figure 8H*), suggesting that the source-sink mechanism much more likely establishes the BMP signaling gradient patterning the zebrafish DV axis.

Finally, we asked whether the source-sink and counter-gradient mechanisms are robust to uniform *chordin* expression. Uniformly expressing *chordin* by mRNA injection into one-cell stage *chordin* mutant embryos has been used to rescue *chordin* mutant embryos to adulthood (*Fisher and Halpern, 1999*), including in our study here. We simulated the system with ubiquitous Chordin production and determined that 426 of the 1452 solutions that fit our WT, *chordin-/-*, and *chordin +/-* data with a BMP diffusivity within 2 um²/s of our measured 4.4 um²/s retain a WT BMP gradient when Chordin is uniformly produced. We did not titrate the Chordin production rate, but had we done so, more simulations may have fit. Interestingly, the 426 remaining solutions are all source-sink mechanisms with gradients of Chordin that are high dorsally and low ventrally (*Figure 8I*).

## Comparing zebrafish and *Drosophila* DV patterning

Our results show that the BMP signaling gradient patterning the zebrafish DV axis is markedly different from the one patterning the *Drosophila* DV axis. The zebrafish BMP signaling gradient is broad, reaching half of its maximum at ~40% of the total DV axis length (*Figure 2F*). In contrast, the *Drosophila* gradient is incredibly steep, reaching half of its peak at only ~10% of the total embryo DV axis length (*Figure 9A*) (*Peluso et al., 2011*; *Sutherland et al., 2003*). Similarly, the loss of the main BMP antagonist in either organism, Chordin or Sog, causes markedly different effects on the BMP signaling gradient (*Figure 9A*) (*Mizutani et al., 2005*; *Peluso et al., 2011*; *Sutherland et al., 2003*).

Zebrafish and *Drosophila* DV patterning differ in both length-scale and time-scale. The *Drosophila* embryo has a 250 μm half-circumference, while the zebrafish embryo has a 700 μm half-circumference. The zebrafish gradient is established gradually in ~2–3 hr and maintained for several hours (*Ramel and Hill, 2013*; *Tucker et al., 2008*), whereas the *Drosophila* BMP signaling gradient is established and patterns DV tissues in ~1 hr (*Dorfman et al., 2001*; *Wang and Ferguson, 2005*). Given these differences, we sought to determine if *Drosophila*-like shuttling simulations could exist with zebrafish time- and length-scales, and if so, how the biophysical parameters of the components would differ from those consistent with the WT and *chordin* mutant P-Smad5 gradients (*Figure 7*).

In the 1,000,000 random simulations tested, we found many simulations that could generate a steep gradient with extensive shuttling in zebrafish. Simulations were considered to be *Drosophila*-like if their WT gradient reached its half-maximum ≤10% of the total embryo circumference and the ventral peak of the *chordin* mutant was 50% lower than the ventral peak level of the WT curve (*Mizutani et al., 2005*). We found 251 simulations that fit the *Drosophila*-like WT and *chordin* mutant signaling gradients. We also excluded simulations with excessive BMP-Noggin interaction, as *Drosophila* does not possess *noggin* homologs (*Figure 9B*).

The *Drosophila*-like simulations required a very mobile Chordin and BMP-Chordin species. *Drosophila*-like simulations required Chordin to have a high range to move to encounter the BMP in the ventral region (*Figure 9C,E*). Similarly, BMP-Chordin needed to have a high range so it could be shuttled a sufficient distance towards the ventral-most region of the embryo (*Figure 9D,E*). The cleavage of free Chordin by Tolloid needed to be low to allow Chordin range to remain high (*Figure 9F*). Conversely, the cleavage of bound Chordin needed to be high to release the BMP from Chordin (*Figure 9F*). Chordin range must be high to allow the formation of a counter-gradient to

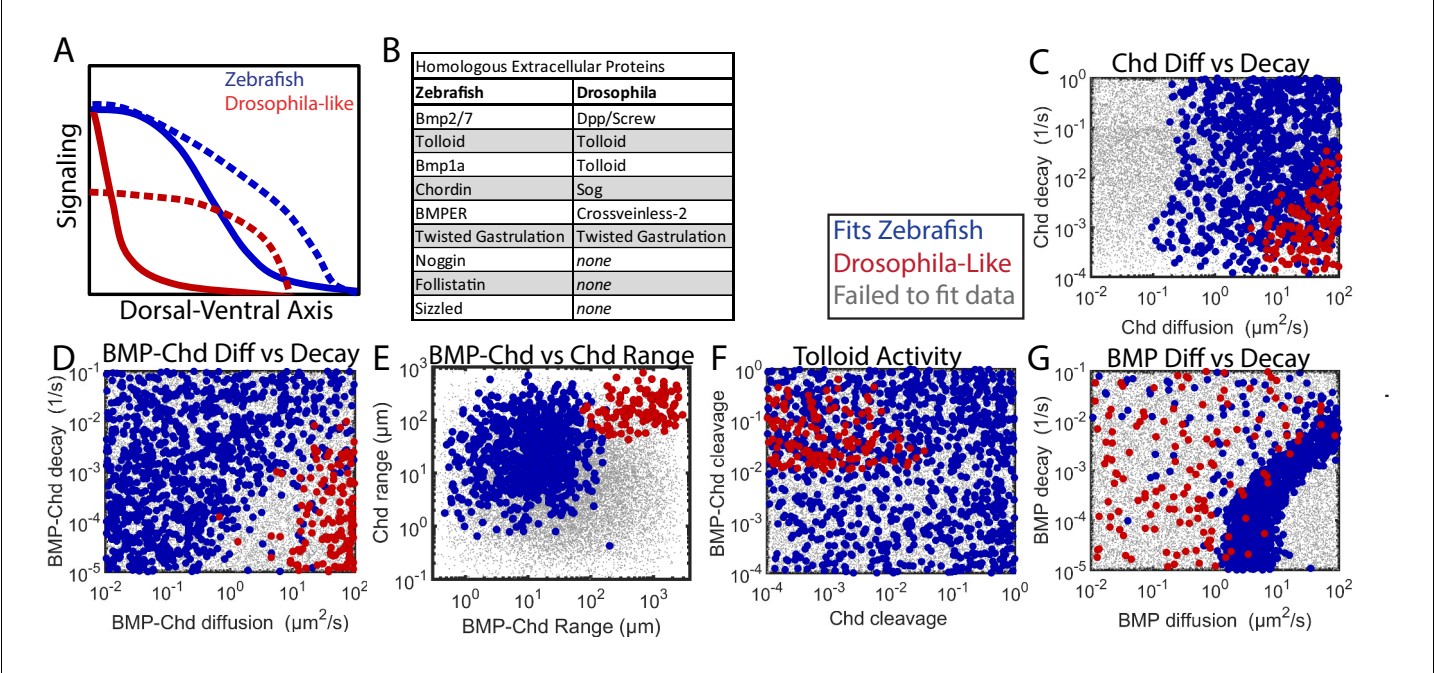

**Figure 9.** Comparing Zebrafish and *Drosophila*-like solutions. (**A**) Depiction of the BMP gradients patterning the *Drosophila* and zebrafish DV axis. The *Drosophila* DV axis has been flipped to match the zebrafish. Solid lines are WT. Dotted lines are *chd* or *sog* LOF. (**B**) List of homologous proteins involved in DV patterning of zebrafish and *Drosophila*. (**C–F**) Solutions able to fit WT and *chd* LOF zebrafish data (blue) vs. solutions capable of fitting *Drosophila*-like WT and *Drosophila*-like *sog* LOF gradients (red). Parameter combinations that failed to fit either are represented as small grey dots. (**C**) Chd diffusivity vs. Chd decay rate, which includes the rate of Chd cleavage by Tld. (**D**) Diffusivity of BMP bound to Chd vs. decay rate of BMP bound to Chd. (**E**) Range was estimated as sqrt(diffusivity/decay). (**F**) Cleavage rate of Chd and BMP-Chd by Tolloid. (**G**) BMP diffusivity vs. BMP decay rate.

DOI: https://doi.org/10.7554/eLife.22199.014

block signaling in the lateral regions of the embryo (*Figure 9E*). Conversely, BMP range was relatively unrestricted in *Drosophila*-like simulations, as the shuttling mechanism relies more on BMP-Chordin mobility than BMP mobility (*Figure 9G*).

## Discussion

Here we have quantified the BMP signaling gradient in WT and *chordin* zebrafish mutants by measuring with high precision the P-Smad5 immunofluorescence level in all ~8,000 + nuclei of an embryo, with high reproducibility within and between embryos at multiple developmental stages. We then used these data to inform a computational model-based screen of over 1,250,000 combinations of biophysical parameters of the major extracellular BMP modulators. We defined mathematical criteria to distinguish between four widely proposed mechanisms to set up the BMP signaling gradient. Our computational model-based screen excludes the shuttling and transcriptional mechanisms as possibilities for establishing our measured WT and *chordin* mutant P-Smad5 profiles, providing compelling evidence that the BMP signaling gradient patterning the zebrafish DV axis is established by either a counter-gradient or source-sink mechanism. We further determined that the effective diffusivity of the BMP ligand in the zebrafish embryo is relatively fast, consistent with 1421 source-sink simulations but only 31 counter-gradient ones (*Figure 8H*). Comparison of models that satisfy zebrafish or *Drosophila*-like gradient profiles suggests that the range of BMP-Chordin differs between zebrafish and *Drosophila*-like DV patterning mechanisms (*Figure 9E*), and the *Drosophila*-like patterning mechanism requires restricted Tolloid degradation rates for BMP-Chordin and Chordin, which were not observed for the zebrafish patterning mechanism (*Figure 9F*).

## Fish vs. flies: A mechanism diverged

The shape of the BMP signaling gradient differs greatly between *Drosophila* and zebrafish DV patterning (*Figure 9A*). The parameters that drive the most significant difference between the *Drosophila*-like and zebrafish simulations are the mobility and processing rates of BMP-Chordin (*Figures 7I* and *9D,F*). For shuttling to be possible in a system with a broad peak of signaling, as it is in zebrafish, the BMP-Chordin cleavage rate needed to be low enough to allow BMP-Chordin to move farther and distribute BMP over a larger region (*Figure 5F,L*). For shuttling to be possible in a system with a tight peak of signaling as it is in *Drosophila*, BMP-Chordin cleavage needed to be rapid to release BMP-Chordin over a smaller region (*Figure 9F*). We show that a shuttling mechanism is not functioning in zebrafish, indicating that this delicate balance of BMP-Chordin mobility and Tolloid cleavage has been lost or did not emerge in vertebrate DV patterning.

Sog and its vertebrate homolog Chordin differ in how they are processed by the metalloprotease Tolloid depending on whether it is bound to BMP ligand. Chordin can be cleaved by Tolloid whether bound to BMP ligand or not (*Piccolo et al., 1997*), while Sog is only cleaved when bound to BMP (*Marqués et al., 1997*). Interestingly, when Sog is mutated to allow it to be processed by Tolloid regardless of BMP binding, the shuttling of BMP-Sog complexes in flies is greatly reduced (*Peluso et al., 2011*), suggesting that this attribute of Sog is necessary for effective shuttling. However, surprisingly, we found numerous shuttling simulations in our zebrafish modeling-based screen with the opposite properties, high Chordin cleavage rates and low BMP-Chordin cleavage rates (*Figure 5F,L*). The requirement for preferential cleavage of BMP-Chordin by Tolloid only emerged when we screened for *Drosophila*-like simulations, in which shuttling was generating a tight peak of BMP signal (*Figure 9A,F*). This suggests that the preferential processing of BMP-Sog by Tolloid seen in *Drosophila* is not an inherent requirement of the shuttling mechanism, but may instead be a result of both the requirement to facilitate shuttling <u>and</u> to generate a steep gradient.

In *Drosophila* DV patterning, the BMP signaling gradient is so steep that its base falls well within the region of *bmp* expression, far from the *sog/chordin* expression domain (*Figure 9A*) (*Francois et al., 1994*; *Holley et al., 1995*). To suppress lateral BMP signaling and form the *Drosophila*-like simulations seen in our mathematical model-based screen (*Figure 9A*), Sog/Chordin needed to have a high range to diffuse far from its site of expression to inhibit BMP signaling over most of the *bmp* expression domain (*Figure 9C,E*). Therefore, the degradation of free Sog/Chordin by Tolloid needed to be low (*Figure 9F*). However, to generate a narrow peak of BMP signaling (*Figure 9A*), BMP-Chordin cleavage by Tolloid needed to be high (*Figure 9F*). Therefore, the requirement to preserve the range of action of free Chordin, combined with the requirement to rapidly cleave BMP-Chordin to generate a steep peak of signaling may explain why the preferential cleavage of BMP-Sog by Tolloid is needed for shuttling in *Drosophila*.

## Comparing the source-sink and counter-gradient mechanisms

While the shuttling mechanism relies on the movement of the bound BMP-Chordin complex, the source-sink and counter-gradient mechanisms rely on the movement of unbound BMP and Chordin. The source-sink mechanism relies on BMP diffusing dorsally to bind Chordin. Conversely, the counter-gradient mechanism relies on Chordin diffusing ventrally to bind BMP. Consistent with this, we found that the majority of simulations consistent with the source-sink mechanism require a high BMP range and a high BMP diffusivity (above 1 $\mu m^2/s$), while the counter-gradient mechanism requires a high Chordin range and high Chordin diffusivity (above 1 $\mu m^2/s$) (*Figures 5* and *7*).

To illustrate the distinct manners by which a source-sink and counter-gradient mechanism would generate the zebrafish BMP signaling gradients observed, we graphically display in *Figure 10* the relative contributions of BMP and Chordin diffusion and BMP-Chordin binding to gradient formation for the median simulations that fit our WT, *chordin* mutant, *chordin* +/-, and Bmp2b-Venus FRAP data. The primary differences between the source-sink and counter-gradient mechanisms manifest in the relative amount of Chordin protein that diffuses ventrally into the *bmp* expression domain and the primary role of Chordin in forming the BMP gradient. A counter-gradient mechanism leads to higher levels of Chordin that extend over a greater region of the ventral *bmp* expression domain compared to a source-sink mechanism (*Figure 10A*). Counter-gradient and source-sink mechanisms also differ significantly in where Chordin binds and inhibits BMP ligand activity along the DV axis, which is consistent with the distinct Chordin protein distributions (*Figure 10A*). In a counter-gradient

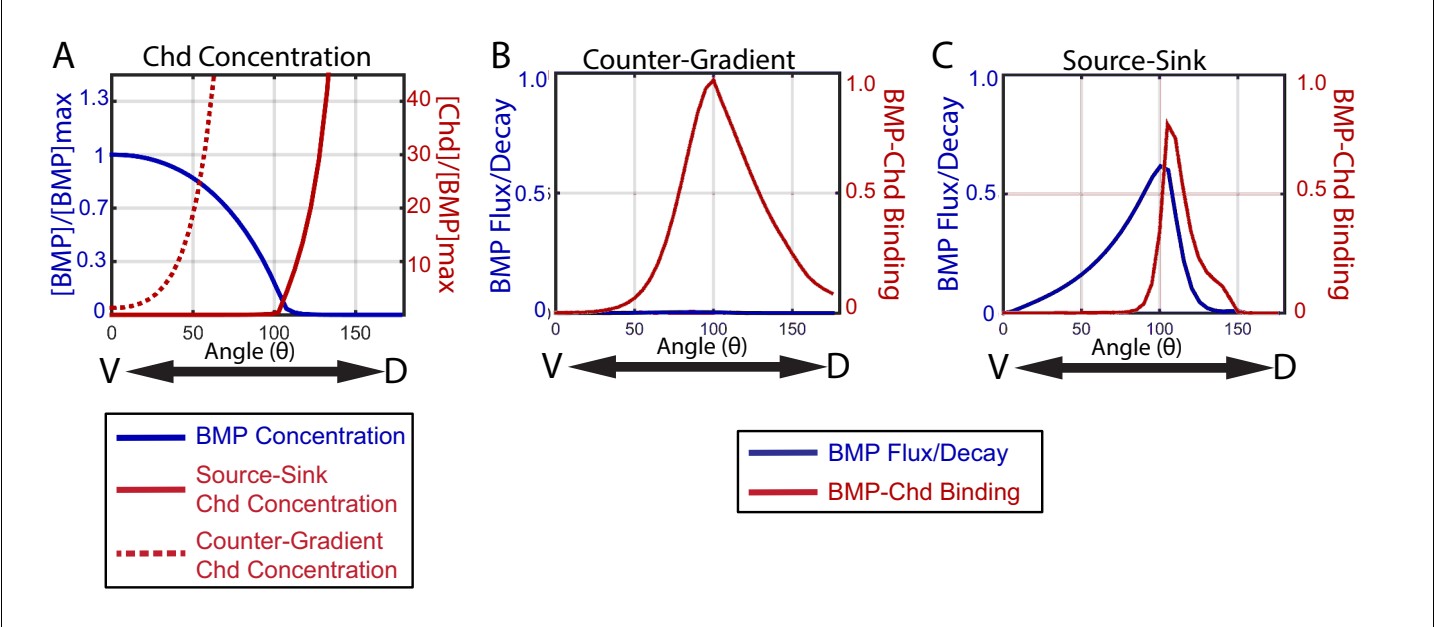

**Figure 10.** How the source-sink and counter-gradient mechanisms shape the gradient. (**A**) The mean BMP and Chd concentrations in all source-sink and counter-gradient solutions fitting WT, *chd* LOF, and *chd* heterozygous P-Smad5 data and within a diffusivity between 2.4 and 6.4 $\mu m^2$/s. (**B**) The diffusive flux divided by the decay [($D_{BMP}$/$dec_{BMP}$)*($d$[BMP]/$dx$)*(1/[BMP]$_{max}$)] of BMP (blue) with units of $10^3$*$\mu m$ and rate of binding of BMP to Chd ($k_{on}$* [BMP]*[Chd]) (red) with units of 3.6*$10^{-2}$*$sec^{-1}$ for representitive (**B**) Counter-Gradient and (**C**) Source-Sink solutions fitting WT, *chd* LOF, and *chd* heterozygous P-Smad5 data and within a diffusivity between 2.4 and 6.4 $\mu m^2$/s (*Figure 8*).

DOI: https://doi.org/10.7554/eLife.22199.015

mechanism Chordin binds BMP in a broader domain, extending over a much greater extent of the DV axis than in a source-sink mechanism (*Figure 10B,C*). In a source-sink mechanism, Chordin binds BMP largely in dorsal regions and extends little ventrally, effectively generating a driving force for the movement of BMP (*Figure 10B,C*). This dorsal sink of Chordin leads to a diffusive flux of BMP ligand down its concentration gradient (*Figure 10C*) that largely shapes the BMP signaling profile in a source-sink mechanism (*Figure 10C*).

Importantly, gradient formation by either of these mechanisms need not be exclusive, and instead characteristics of each can contribute to some extent in shaping sectors of the other's gradient. In the source-sink simulation in *Figure 10A,C*, Chordin forms a small counter-gradient partially contributing to the gradient shape in this region. Thus, in some source-sink simulations, Chordin shaped the gradient by simultaneously binding ligand to block signaling in lateral positions and by establishing a sink that serves as a driving force for the diffusive flux of ligand from ventral positions dorsally towards the regions of higher Chordin. Though each solution is classified by the dominant mechanism, many share some aspects of both the source-sink and counter-gradient mechanisms.

## BMP diffuses relatively freely

We measured BMP diffusivity for the first time in vertebrates. Using FRAP, we show that BMP can diffuse relatively freely with a diffusivity of 4.4 ± 0.4 $\mu m^2$/s (*Figure 8E*), about 4-fold less than secreted Venus diffusion in the zebrafish blastula ($\approx$ 16 $\mu m^2$/s) (*Figure 8G*). Our measured BMP diffusivity is comparable to the diffusivity of Squint (Ndr1, D = 3.2 $\mu m^2$/s), another TGF-$\beta$ ligand in the zebrafish blastula that acts as a long-range mesoderm inducer (*Müller et al., 2012*). This high BMP diffusivity is consistent with previous BMP heterodimer protein injections into the extracellular space of BMP-deficient embryos, which could extend throughout the embryo and restore the WT P-Smad5 gradient within 1.5 hr, suggesting the BMP ligand can move rapidly (*Little and Mullins, 2009*).

Our computational screen found hundreds of simulations with a BMP diffusivity near 4.4 $\mu m^2$/s. The vast majority of those simulations were classified as having a source-sink mechanism. The remaining few are classified as having a counter-gradient mechanism. This paucity of counter-

gradient simulations is a reflection of the fine-tuning needed for this mechanism to work as compared to the source-sink mechanism. The counter-gradient mechanism requires a specific balance of Chordin diffusivity and decay as well as Tolloid degradation, while the source-sink mechanism does not (*Figure 7H,I*). Together, this suggests that the source-sink mechanism is more robust to changes in biophysical parameters than the counter-gradient mechanism, but does not rule out the counter-gradient mechanism entirely.

## BMP transcriptional feedback: A symptom not a cause

The recent observation that *bmp2b* and *bmp7a* are expressed in a graded manner in WT embryos has lead to the hypothesis that the BMP signaling gradient may largely reflect the *bmp* expression gradient (*Ramel and Hill, 2013*). We sought to determine if the BMP signaling gradient in zebrafish is predominantly established by matching the *bmp* expression gradient. Three results do not support this hypothesis. First, the relative shape of the *bmp2b* expression domain (*Figure 3*) did not precisely match the P-Smad5 gradient (*Figure 2*). Second, we mathematically defined a gradient established by a transcriptional mechanism as one where 80% of the BMP accumulates or is degraded where it is produced, as opposed to binding antagonists or diffusing away. When we performed a mathematical model-based computational screen using a *bmp* expression profile that reflected the P-Smad5 WT gradient, no tested parameter combination fit both our measured WT and *chordin* LOF P-Smad5 gradients. For the transcriptional mechanism to work, the *bmp* expression domain would have to change in the *chordin* mutant condition to fit the mutant gradient profile.

Finally, we showed that feedback by BMP signaling on *bmp* expression does not begin until after 5.7 hpf. This is likely because the initial *bmp* expression domain is established by maternal factors and repression from Bozozok, a transcription factor activated by maternal Wnt signaling (*Koos and Ho, 1999*; *Langdon and Mullins, 2011*; *Leung et al., 2003*; *Solnica-Krezel et al., 2001*). FGF (Fibroblast Growth Factor) and Nodal signaling also repress *bmp* expression dorsally (*Fürthauer et al., 2004*; *Kuo et al., 2013*; *Maegawa et al., 2006*; *Shimizu et al., 2000*; *Varga et al., 2007*). Importantly, BMP signaling does not play a role in the initial establishment of the *bmp* and *chordin* expression domains at 4 hpf, as both are unchanged prior to ~6 hpf in BMP pathway mutants (*Kishimoto et al., 1997*; *Miller-Bertoglio et al., 1997*; *Schmid et al., 2000*). At or after ~6 hpf, the *bmp* expression domain begins to respond to changes in BMP signaling levels, as *bmp* expression decreases in BMP pathway mutants and increases in BMP antagonist mutants (*Kishimoto et al., 1997*; *Miller-Bertoglio et al., 1999*; *Nguyen et al., 1998*; *Ramel and Hill, 2013*; *Schmid et al., 2000*). We show that the *bmp2b* expression domain does not begin to shift in response to the loss of *chordin* until after 5.7 hpf (*Figure 3*), while the P-Smad5 gradient shifts as early as 4.7 hpf (*Figure 6*).

While we show that the transcriptional mechanism cannot entirely account for the measured P-Smad5 WT, *chordin* +/-, and *chordin* -/- profiles, the shape of the transcriptional gradient does still contribute to gradient shape. Four-fold more individual solutions fit the WT, *chd* +/-, and *chd* -/- P-Smad5 profiles when the *bmp* expression input matched our measured *bmp2b* level (0.4%) than when *bmp* expression was set to match the P-Smad5 profile (0.1%) (*Figure 7E,K*). Counter-gradient solutions only appeared when the *bmp* expression input was broader than the P-Smad5 profile (*Figure 7E',K'*). More source-sink solutions were present when the *bmp* expression input was broader because BMP did not need to travel as far or fast (*Figure 7F,G* vs. *Figure 7L,M*) to reach the dorsal sink. Together, these effects show that the shape of the *bmp* expression domain contributes to BMP gradient formation even if it does not entirely account for it, similar to that found for the *bicoid* transcript and protein gradients that pattern the Drosophila AP axis (*Little et al., 2011*).

## Integrated approach reveals source-sink mechanism

Although we and most others in the vertebrate field have contended that a Chordin (or BMP antagonist) counter-gradient drives formation of the BMP activity gradient in DV patterning, our studies here, intriguingly, suggest that an alternate source-sink mechanism may prevail. While the source-sink gradient mechanism is also modulated by Chordin, Chordin instead acts in a distinct manner as a sink, binding BMP ligand predominantly in dorsal regions, thus allowing a BMP diffusive gradient to form throughout most of the ventral half of the embryo. Key to deriving this alternate model was the integrated approach used that combined quantitative experimental analysis with computational

modeling. Importantly, a role for Chordin in establishing a sink that drives gradient formation would not have been revealed to us had we not performed the mathematical model-based computational screen. By narrowing the compatible simulations successively with the WT, *chordin +/-*, and then *chordin* mutant P-Smad5 profiles, 15-fold more source-sink simulations than counter-gradient simulations perdured (*Figure 7E'*). Furthermore, the computational modeling also illuminated the BMP diffusivity parameter as one to further test the source-sink mechanism. Significantly, our measured Bmp2 diffusivity further supports the source-sink mechanism of gradient formation. Thus the seamless integration of quantitative experimental analysis with mathematical model-based computational screens has proved to be a highly successful approach to elucidating mechanisms of BMP gradient formation. Future studies will be required to definitively determine the mechanism and further test the source-sink and counter gradient models of BMP gradient formation.

## Materials and methods

### Zebrafish lines

WT TU (RRID:ZIRC_ZL57) zebrafish were used as the wild-type strain. Adult *chd*^tt250/tt250^ (RRID:ZFIN_ZDB-GENO-060811-12) or *chd*^tt250/Tu^ zebrafish were used to produce *chd* homozygous and heterozygous embryos. Adult *chd*^tt250/tt250^ homozygotes were generated by injecting *chordin* mRNA into 1-cell stage *chd-/-* embryos to rescue the ventralization phenotype and then were raised to adulthood. Genotyping of adults and embryos was performed using KASPar genotyping (*Smith and Maughan, 2015*) with primers designed and generated by KBioscience of the following sequences, where [G/A] is the WT/mutant nucleotide:

CTCCTTCGGTGGCCGCTTTTACTCTCTGGAAGACACGTGGCATCCAGATCTCGGAGAGCCG
TTTGGTGTGATGCACTGCGTTATGTGTCATTGTGAGCCG[G/A]TGAGTTGTGCACAGTTCAG
TTTGAAATCCATATTGAATCTGAATTGACTTCTGCTGCTGAGTTGCAACATTCACACCATATCTAAA
TTGAATTCATATT

### mRNA injection for BMP overexpression

WT zebrafish embryos were injected with mRNA at the 1-cell stage. *bmp2b* and *bmp7a* RNA were made using the SP6 MMessage Machine kit (Life Technologies AM1340). *bmp2b* or *bmp7a* cDNA in a pBluescript II KS-construct was linearized with NotI. To overexpress BMP, 6 pg of *bmp7a* RNA and 12 pg of *bmp2b* RNA were injected. Resulting embryos had a V5 fully ventralized phenotype at 24 hpf (*Figure 6H,I*).

### In situ hybridization and domain size analysis

Whole-mount in situ hybridizations were performed using RNA DIG probes as described using, *chordin* (*Miller-Bertoglio et al., 1997*) and *noggin1* (*Dal-Pra et al., 2006*). RNA probes were generated using the Roche DIG RNA labeling kit (11277073910). Embryos were cleared in glycerol, and photographed using a Leica IC80 HD. Images were processed using ImageJ and MATLAB. In situs were stained with Anti-DIG-Alkaline Phosphatase (Roche 11093274910) and developed using BM Purple (Roche Life Sciences).

The sizes of the *chordin* and *noggin* expression domains were determined by image processing with MATLAB. Centerpoints of animal views of each embryo were determined by thresholding. The boundaries of the *noggin* and *chordin* expression domains relative to the center of the animal view were determined by a second threshold. These points were connected by line segments and the angle was measured (*Figure 3A,B*).

### Fluorescent *bmp2b in situ* hybridization and image analysis

Fluorescence in situ hybridization (FISH) was performed on fixed cryosections using a RNAscope Fluorescent Multiplex Kit (Advanced Cell Diagnostics (ACD)). Embryos were fixed with 4% paraformaldehyde in PBS at 4°C overnight. Embryos equilibrate in 30% sucrose until they sink and incubated in fresh 30% sucrose for 3 days at 4°C. Cryosections (20 μm) at the marginal region were collected on slides, followed by air drying for 30 min at −20°C. In situ hybridization was performed according to the manufacturer's instructions (ACD). A custom C2 probe was designed for *bmp2b*

(#456471-C2). *bmp2b* probe was used at 1:10 dilution. Sections were stained for DAPI and images were acquired at 63 × oil objective using a Zeiss 800 upright confocal.

Relative intensity quantification of mRNA levels was performed on maximum intensity projections of 20 μm sections. Marginal cells were grouped into 10 degree intervals along the marginal circumference. Average intensity was quantified in each section using the MATLAB image analysis toolbox. Averaged *bmp2b* mRNA levels in 2.5 hpf embryos were used to measure background. We found equivalent intensity levels and distributions in the 2.5 hpf embyros and the dorsal *bmp2b* signal in Wt 5.7 hpf embryo suggesting limited to zero *bmp2b* expression in the dorsal region. For each cross-section, the right and left side of the distributions were averaged into a single ventral to dorsal profile.

## Immunostaining

Embryos were fixed overnight in 4% paraformaldehyde at 4°C, blocked in NCS-PBST (10% fetal bovine serum, 1% DMSO, 0.1% Tween 20 in PBS), and probed overnight with a 1:100 dilution of anti-phosphoSmad1/5/8 antibody (Cell Signaling Technology, #9511, discontinued), followed by a 1:500 dilution of goat anti-rabbit Alexa Fluor 647-conjugated antibody (Thermo Fisher Scientific, Rockford, IL; Cat# A-21244, RRID:AB_2535812). Embryos were mounted in BABB (benzyl alcohol (Sigma B-1042) and benzyl benzoate (Sigma B-6630), 1:2 ratio) and scanned using a Zeiss LSM 710 confocal microscope with a LD LCI Plan-Achromat 25x/0.8 Imm Corr DIC M27 multi-immersion lens. The oil-immersion setting was used to reduce Mie scattering distortion, spherical aberrations, and chromatic aberrations by minimizing refractive index (R.I.) mismatch between the lens oil (R.I. = 1.518), the coverslip, BABB (R.I. ≈ 1.56), and the light scattering particles in the embryo (R.I. ≈ 1.56). Fluorophore bleaching was greatly reduced by precise embryo orientation, reducing sample thickness, and by high scan speeds using a Zeiss LSM 710 confocal microscope. Nuclei were visualized with Sytox Orange (Molecular Probes) or Sytox Green (Molecular Probes).

## Summary of imaging and processing

Immunostained P-Smad5 embryos were processed and imaged as described above. We observed minimal photo-bleaching and spherical aberration (*Figure 11A–C*). We wrote a Matlab algorithm capable of identifying all 8000 + nuclei centerpoints in each embryo in 3D, removing populations unresponsive to P-Smad5 (such as yolk syncytial nuclei and dividing cells), and extracting the P-Smad5 intensities associated with each nucleus (*Figure 11D–L*). The resulting individual digital embryos (*Figure 2A'–E'*) from each condition were averaged together to generate large datasets from which embryo-wide P-Smad5 levels could be quantified in WT and mutant conditions (*Figure 6A–B*).

## Image processing

Nuclear intensities of P-Smad5 were extracted from the stacks of images generated using Matlab algorithms (source code in *Supplementary files 1*).

The centerpoints of all the nuclei were located using the Sytox DNA stain. The '.lsm' files were converted to '.tif' files using ImageJ, and then imported into Matlab as 1024 × 1024 X Z multidimensional arrays. XY pixels were 0.55 um, Z pixels were 2.3 um. The images were then smoothed using a 9 × 9×3 kernel (most nuclei are 10 × 10×4 pixels large). Local minima and maxima were removed using the 'imhmax' and 'imhmin' functions. The remaining maxima were found using the 'imregionalmax' function on the entire 1024 × 1024 X Z array. Maxima closer together than six pixels were assumed to be in the same nucleus, and were combined. The remaining maxima were assumed to be the centerpoints of the nuclei (*Figure 11E*).

These centerpoints were used to extract P-Smad5 intensities on the P-Smad5 channel. P-Smad5 distribution in each nucleus was approximately uniform, so a small sphere within each nucleus was averaged to attain the P-Smad5 intensity. On the P-Smad5 channel, pixels within a spherical 6 × 6 × 3 kernel of each maxima were averaged.

Cell types unresponsive to Bmp signaling were removed. P-Smad5 appears to be uniformly distributed throughout the cytoplasm during cell division, making measurement impractical (*Figure 11G*). In dividing cells, chromatin condenses making DNA stains such as Sytox concentrated and bright. Cells with a bright DNA fluorescence staining above 140% of the mean DNA

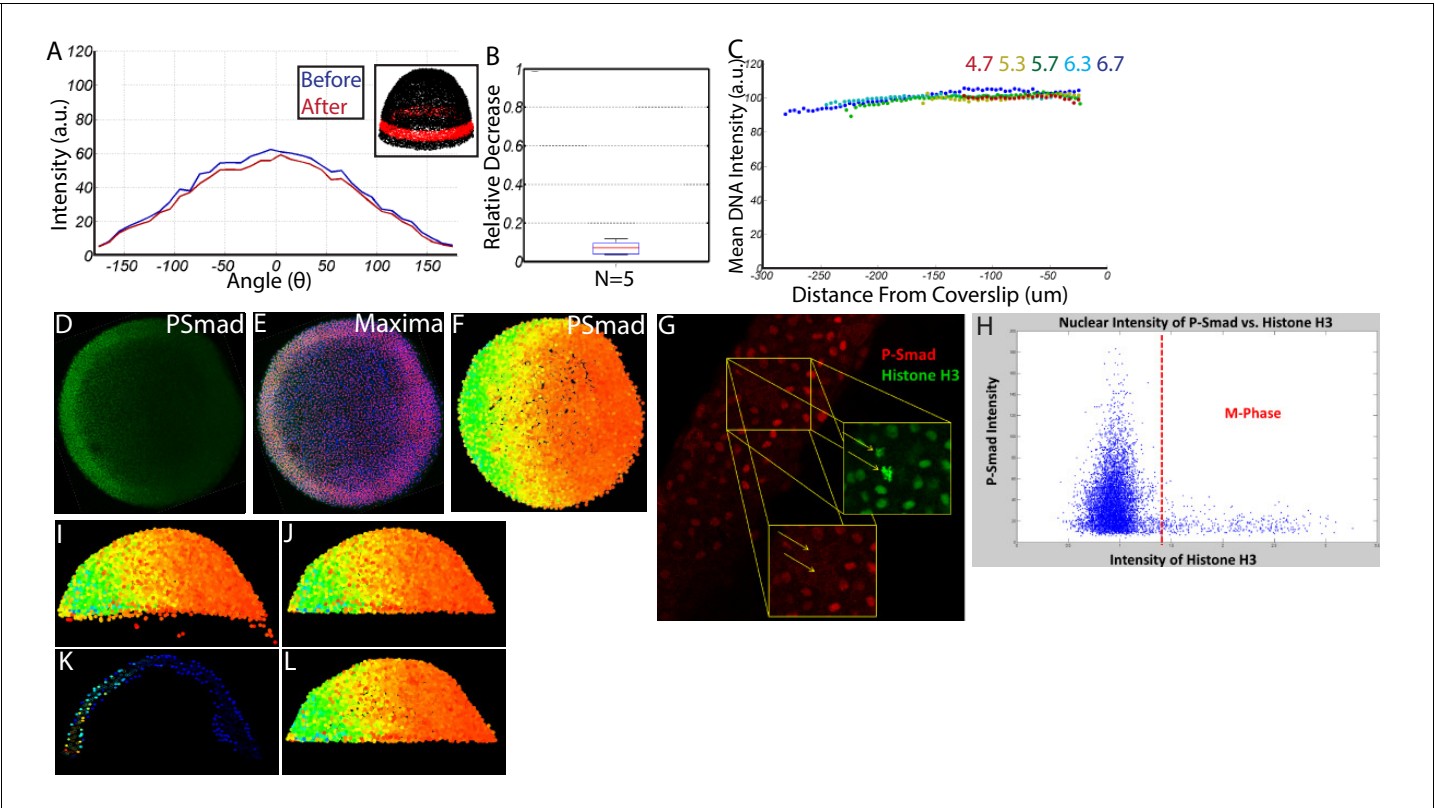

**Figure 11.** Quantifying nuclear P-Smad5 intensities embryo-wide. (**A**) Marginal P-Smad5 intensity from a *chd* LOF embryo imaged twice. (**B**) Average P-Smad5 intensity drop-off from photo-bleaching of all nuclei in embryos imaged twice (N = 5). (**C**) There is minimal intensity drop-off due to spherical aberration, as shown by the average intensity of the nuclear DNA stain (Sytox Orange) versus distance from the coverslip (4.7: N = 3, 5.3: N = 4, 5.7: N = 13, 6.3: N = 11, 6.7: N = 4). (**D**) Maximum projection of an animal view of a single embryo. (**E**) Nuclei centerpoints (red dots) identified from the sytox nuclear stain (blue). (**F**) Measured centerpoint nuclear intensities displayed as a heatmap. (**G**) P-Smad5 is absent in dividing cells (red stain, yellow arrows). Dividing cells have bright condensed chromatin (green stain, yellow arrows). (**H**) Bright condensed chromatin was used to identify dividing cells. Cells with a 40% elevated DNA stain over the mean (red line) were eliminated from the analysis. (**I**) Lateral view of a single embryo. (**J**) Sparse Yolk Syncytial Layer nuclei below the margin are eliminated. (**K**) Single lateral slice depicting the elimination of remaining yolk syncytial layer nuclei and enveloping layer nuclei by subtracting the outer 15% of all nuclei (filled in circles) to leave only deep cell nuclei (open circles). (**L**) Lateral view of embryo after outer 15% has been eliminated.

DOI: https://doi.org/10.7554/eLife.22199.016

fluorescence were considered dividing and eliminated from the analysis. Extra-embryonic cells such as the Enveloping Layer (EVL) and the Yolk Syncytial Layer (YSL) did not appear to respond to BMP ligand the same way as the deep cells. These cell types are not patterned along the DV axis by BMP, and were eliminated from our analysis. To do so, sparse EVL cells located below the vegetal margin were eliminated by hand (*Figure 11J*). Next, the inner and outer layer of approximately 15% of the total cells was eliminated (*Figure 11K,L*). The remaining cells were assumed to be non-dividing deep cells.

Embryos of similar stages were then aligned and conformed to a template embryo of the same stage using Coherent Point Drift (CPD) (*Myronenko and Song, 2010b*). Embryos were aligned in the AP direction by fitting them to a sphere, finding a plane that spanned the marginal region, and rotating until that plane was aligned with the XY axis. Embryos were aligned in the DV direction using the embryonic shield as a morphological marker. Before 6 hpf, when the shield is not present, the embryos were aligned in the DV direction by fitting a polynomial regression to the P-Smad5 gradient around the margin and rotating until the max peak was ventral. Next, embryos were all aligned to a template using an affine CPD (*Myronenko and Song, 2010b*). This corrected for any distortions in embryo shape that may have occurred during fixation and staining.

Embryos from the same set were subjected to no normalization. Embryos stained and imaged on different days with different settings were normalized by multiplying the entire set by a single scalar value. To determine this normalization scalar, control WT embryos were always imaged in conjunction with each experimental condition. The scalar normalization value was determined by minimizing the sum of the error between the control WT embryos imaged on different days.

## Generating and comparing P-Smad5 profiles around the margin of the embryo

To generate marginal profiles, a 40 um thick band of cells around the vegetal margin was chosen for each embryo. Cells within that band were grouped into 10 degree intervals and averaged together to form 36 individual points. The left and right side of the gradient were averaged together into a single ventral to dorsal profile. For 3-D embryo-wide averages, all nuclei were projected onto a sphere fitting the embryo. The sphere was then divided into 4800 approximately equilateral triangles. All nuclei falling within each triangle were averaged together. Slopes were obtained by fitting a lowess fit to the averaged 3-D data's spherical coordinates phi and theta using the 'fit' function in Matlab. To determine if the marginal gradients of WT and *chordin* mutant embryos were significantly different, two-tailed T-Tests were performed with a rejection of the null hypothesis at the 5% significance level (*Figure 6*, *Figure 6—figure supplement 1A*).We observed a difference in WT vs. *chordin* mutant embryos that was much larger than our observed embryo-to-embryo variability (*Figure 6*). Our t-tests confirmed that our sample sizes are sufficient to discern differences between the WT and *chordin* mutant embryos (*Figure 6*, *Figure 6—figure supplement 1A*).

## Mathematical model-based computational screen method

For each set of parameters defined in the parameter vector, we solved the five non-linear partial differential equations (PDEs) for BMP ligand, Chordin, Noggin, and the complexes of BMP-Chordin, BMP-Noggin in MATLAB (*Figure 4*, *Figure 4—figure supplement 1A*). Equations were solved on the half-circumference, with 'symmetry' boundary conditions imposed on the first and last nodepoint in the spatial discretization. The half-circumference was discretized into 36 node points with equidistant spacing and the second order spatial derivative is discretized via the finite difference method. The production regions of BMP ligand, Chordin, Noggin, are specified along the nodes by mapping the spatial position to subsequent node position (*Figures 3* and *4D,E*, *Table 1*). Tld is treated parametrically as a function of position according to its domain of expression (*Figure 4D,E*, *Table 1*). Time-stepping of the simulation is handled by the adaptive solver ode15s with a relative tolerance set to 1e-9. The model is solved for the developmental window that spans from 3.5 to 5.7 hpf and all measurements of model error are calculated at 5.7 hpf.

For each model iteration, parameters were selected from a uniform distribution in log space that covered four orders of magnitude within the physiological range for each parameter. Each parameter was selected independently of the remaining parameters. Adaptive and subsampling methods that increase parameter selection in regions with high variance in model output were not used in our parameter selection for a number of reasons. Firstly, a parameter matrix is produced that is then subdivided across distributed computers to solve the PDEs to produce a stored file of model solutions and parameter vectors associated with the stored solution. For each parameter vector, the PDE system is converted into a set of 180 ordinary differential equations after discretization and dynamically solved for WT, *chd−/−*, *nog−/−*, and *chd+/-* conditions using the implicit solver ode15s. Ode15s is well suited to problems with numerical stiffness that arise during numerical screens with random parameters. Thus, for an ensemble of 1 million parameter vectors, the system of differential equations are solved four times to simulate the wt and mutant conditions, increasing the total model evaluations to 4 million. Following calculation of the model solutions for each parameter vector, post processing, sorting, and calculation of model fitness against the data is handled by a separate program that operates on the stored solutions. The separation of model evaluation from model analysis allows for much greater flexibility, the total number of simulation results, and an ability to add additional simulation results to existing simulations that have previously been computed. If a job is interrupted, the index and solutions up to the parameter index are stored and simulation jobs can be restarted at the last iteration point.

Random sampling is not susceptible to the 'curse of dimensionality' that is common to regular grid spacing methods allowing for more efficient estimation of the NRMSD landscape in *Figure 7(C, D)*, and random points fill in the parameter space more evenly than regular grid spacing that forces evaluations of the model within the same parameter hyperplane (*Caflisch et al., 1998*). *Figure 7C,D* demonstrates the dense coverage and sampling to estimate the NRMSD values and dense coverage in regions with acceptable NRMSD values. Random sampling of the biophysical parameters is common to these types of problems (*Eldar et al., 2002*; *von Dassow et al., 2000*).

For each parameter vector, the model is initially solved against WT conditions and subsequent simulations for mutant conditions are carried out for the same parameter vector by setting the corresponding production rates for the mutant to zero and re-simulating. Error between the model results and the fluorescent data are calculated via a two-step process. First, the amplitude of the P-Smad5 fluorescent-intensity data and model peak levels for free BMP are normalized as commonly done when calculating a residual with fluorescent intensity data (*Hengenius et al., 2014*; *Pargett and Umulis, 2013*). This approximation is valid considering that (1) BMP ligands are not saturating receptors and (2) Smad5 activity is not saturated (*Figure 6G*). The scaling parameter determined for model-fitness against P-Smad5 was then applied to the remaining model results to capture any changes in BMP levels in the mutant simulations. Residuals were calculated for WT and mutant conditions independently and simulations were scored for passing the WT and mutant conditions independently as opposed to using an aggregate residual. Simulations were classified as transcriptional, source-sink, counter-gradient, or shuttling.

## Constructs, mRNA, and morpholinos (MOs) for FRAP

Sequence encoding fluorescent protein Venus was amplified from pBSK12-*her1:Ub2-Venus* (a gift from Sharon L. Amacher, Ohio State University, OH) (*Delaune et al., 2012*). This sequence was inserted between the pro- and mature domains of Bmp2b, two amino acids downstream of the pro-protein convertase (PC) cleavage site (REKR) with a GSTGTTGGG linker separating the prodomain and the fluorescent protein and a GS linker (GGGGSGGGGS) separating the fluorescent protein from the mature domain. This fusion construct was modified from pCS2(+)-HA-Bmp2b (*Little and Mullins, 2009*). Sequences encoding Venus protein were also fused to the pro-domain of Bmp2b two amino acids downstream of the proprotein convertase (PC) cleavage site (REKR) with a GSTGTTGGG linker, to generate the secreted-Venus plasmid that lacks the mature Bmp2b ligand domain.

Capped mRNA was synthesized using the mMessage mMachine Kit (Ambion) with SP6 RNA polymerase according to the manufacturer's protocol. Vectors were linearized by digestion with NotI. mRNA encoding the Bmp2b-Venus or secreted-Venus, was injected into one- or eight-cell stage embryos. For rescue experiments, 1 ng of *bmp2b* MO2 (GTCTGCGTTCCCGTCGTCTCCTAAG) was injected along with 9 pg of *bmp2b-venus* RNA from a different needle (*Imai and Talbot, 2001*). To perform FRAP in the absence of Chordin, we injected embryos at the 1 cell stage with 1 ng of *chordin* MO1 (ATCCACAGCAGCCCCTCCATCATCC) (*Nasevicius and Ekker, 2000*). Next, *bmp2b-venus* RNA was injected into a single blastomere at the 8-cell stage. Associated phenotypes are shown in *Figure 8B*.

## Western blot

Zebrafish embryos were lysed in Pierce RIPA buffer (89900, Thermo Scientific) supplemented with Halt Protease Inhibitor Cocktail (1862209, Thermo Scientific) and Phosphatase inhibitor Cocktail (1862495, Thermo Scientific). Protein samples mixed with Laemmli sample buffer (Bio-rad) were denatured by incubation for 5 min at 98°, and resolved by SDS-PAGE using Mini-PROTEAN TGX Gels (10%, Bio-rad) and transferred to PVDF membranes (Bio-rad). The membranes were blocked with 5% non-fat milk (Bio-rad) in PBST for 1 hr at room temperature, and incubated with primary antibodies in 2% BSA (Sigma) in PBST at 4°C overnight. After that, the membranes were incubated with HRP-coupled secondary antibodies for 1 hr at room temperature. Chemiluminescence was detected using Clarity Western ECL Substrate (Bio-rad) to obtain the image. Using stripping buffer (46430, Thermo Scientific), the membranes were reused to detect β-Actin as loading controls.

Fluorescence Recovery After Photobleaching (FRAP) mRNA encoding the Bmp2b-Venus fusion protein (50 pg) was injected at the one-cell stage to test the activity of the mRNA in a ventralization

assay. Embryos used for FRAP were injected in one cell at the 8-cell stage. Embryos were mounted in 1% low melting point agarose (Sigma) in glass bottom microwell dishes (MatTek Corporation). FRAP experiments were performed using a LSM 800 confocal microscope (Zeiss) with a W Plan-Apochromat 20×/1.0 objective (D = 0.17 M27 75 mm). Photobleaching in a square region (160.4 µm × 160.4 µm) was performed through the depth of the blastoderm with 100% laser power for ~10 min. In the secreted Venus FRAP (*Figure 8D*), some recovery occurred at the t = 0 time point at the periphery of the bleached region. Recovery of fluorescence was monitored every 10 s in the same imaging plane.

## Processing of FRAP data

From the 8-cell stage injected embryos, regions lacking Bmp2b-Venus producing cells (visualized by high intensity signal throughout the cytoplasm) were identified. Cells displayed characteristic higher intensity signals in the intercellular space and no signal was detected intracellularly in the non-producing cells. Images are taken before the FRAP experiment commences, and saved every 10 s during acquisition. All files are exported in lossless TIFF format for subsequent quantification in MATLAB. To measure the recovery, all TIFF files are imported into MATLAB and the FRAP region is identified for subsequent measurements. Internal FRAP region is scaled from 8-bit [0 255] to [0, 1] and an extracellular mask is generated by removing background with a minimum threshold level set at 1% of the image maximum value. This excludes the intracellular compartments from biasing the average intensity calculations for the extracellular recovery. With background removed, recovery is calculated as the average fluorescence intensity of the extracellular fluorescence within the masked region.

## Calculation of diffusion coefficients from FRAP data

The FRAP region is modeled using a finite-difference equation for diffusion in the FRAP region. Diffusion was estimated by measuring model recovery starting from zero initial conditions and constant concentration boundary conditions. Masked region dimensions for measurement were mapped directly to node-points and distances in the finite difference model to compare and optimize recovery in the mask region. A steepest-descent optimizer with multiple starts was used to estimate the diffusion coefficient for each FRAP experiment. We did not explicitly model the occlusion and tortuosity of the diffusion process caused by the arrangement of cells, nor did it account explicitly for binding and unbinding to HSPGs and other immobile binding components. The impact of binding and the role of occlusions in the diffusion path are very well known (*Cussler, 2009*; *Garnett, 1904*). Therefore our measured diffusion coefficients are the effective diffusion coefficients in zebrafish and not an intrinsic measurement of the diffusion coefficient in a free environment.

## Experimental replicate information

All listed replicates in the figure legends are biological replicates.

### Whole-mount in situ hybridization

For *chordin* and *noggin* domain size, all biological replicates are reported in *Figure 3C*, while representative images are shown in *Figure 3A–B*. All biological replicates were from a single cross stained with different probes. All embryos were reported with no elimination of outliers. For *bmp2b* expression profile, representative images are displayed for each condition and developmental stage. Biological replicates were taken from two experiments.

### Immunohistochemistry

All biological replicates are indicated and averaged in *Figure 2* and *Figure 6* and associated legends. Biological replicates from the 4.7 and 5.3 hpf timepoints for the WT and *chordin* mutant embryos in *Figure 2* and *Figure 6* were from a single experiment. The 5.7 and 6.3 hpf timepoints were collected from 3 and 4 experiments respectively. The 6.7 hpf (WT-only) timepoint was collected from two experiments. In all cases, only embryos that were clearly damaged during the process were eliminated from the analysis. Mean intensities at each point were an average of a subset of the 8000 + nuclei present in each embryo. The XYZ coordinates and P-Smad5 values of all nuclei in all embryos used is attached as a supplemental file.

## FRAP

All biological replicates are shown in *Figure 8E–G*. No outliers were eliminated.

### Supplemental data files

The 'Image Analysis' folder in *Supplementary file 1* contains all files used to identify nuclei, align, and extract P-Smad5 intensities from the '.lsm' files generated by confocal imaging. A Read-Me is included with details on how to use the scripts. A conceptual description can be found above in the materials and methods section. All raw XYZ coordinates and P-Smad5 intensities are included in the 'P-Smad Intensities' folder. Each embryo is represented as a five by n array, where columns 1–3 are XYZ coordinates, column four is nuclear stain intensity, and column five is P-Smad5 intensity.

# Acknowledgements

We thank the UPenn Microscopy Core, especially Andrea Stout and Xinyu Zhao; undergraduate students Samantha Warrick and Dan Zhao; funding from NIH grants T32 HD08318, R01GM056326, R01HD073156, and an NSF Graduate Fellowship; editing help from Francesca Tuazon; and Caroline Hill, Sharon Amacher, Bernard Thisse, and Christine Thisse for reagents.

# Additional information

### Funding

| Funder | Grant reference number | Author |
| --- | --- | --- |
| National Institute of General Medical Sciences | R01GM056326 | Joseph Zinski<br>Mary C Mullins |
| Eunice Kennedy Shriver National Institute of Child Health and Human Development | R01HD073156,T32 HD08318 | Joseph Zinski<br>Ye Bu<br>Xu Wang<br>Wei Dou<br>David Umulis<br>Mary C Mullins |
| National Science Foundation | | Joseph Zinski |

The funders had no role in study design, data collection and interpretation, or the decision to submit the work for publication.

### Author contributions

Joseph Zinski, Conceptualization, Data curation, Software, Formal analysis, Funding acquisition, Validation, Investigation, Visualization, Methodology, Writing—original draft, Writing—review and editing; Ye Bu, Xu Wang, Data curation, Formal analysis, Validation, Investigation, Visualization, Methodology, Writing—original draft, Writing—review and editing; Wei Dou, Software, Visualization, Methodology; David Umulis, Conceptualization, Data curation, Software, Formal analysis, Supervision, Funding acquisition, Validation, Investigation, Visualization, Methodology, Writing—original draft, Project administration, Writing—review and editing; Mary C Mullins, Conceptualization, Formal analysis, Supervision, Funding acquisition, Validation, Investigation, Visualization, Methodology, Writing—original draft, Project administration, Writing—review and editing

### Author ORCIDs

Joseph Zinski, http://orcid.org/0000-0002-5120-5386
David Umulis, http://orcid.org/0000-0003-1913-2284
Mary C Mullins, https://orcid.org/0000-0002-9979-1564

### Ethics

Animal experimentation: This study was performed in strict accordance with the recommendations in the Guide for the Care and Use of Laboratory Animals of the National Institutes of Health. All of the animals were handled according to approved institutional animal care and use committee

(IACUC) protocols (#803105, #804931) of the University of Pennsylvania Perelman School of Medicine.

## Decision letter and Author response

Decision letter https://doi.org/10.7554/eLife.22199.019
Author response https://doi.org/10.7554/eLife.22199.020

## Additional files

### Supplementary files

• Supplementary file 1. Image analysis and imaging data.
DOI: https://doi.org/10.7554/eLife.22199.017

• Transparent reporting form
DOI: https://doi.org/10.7554/eLife.22199.018

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
