## [Decision Letter]

Thank you for submitting your article "Systems biology derived source-sink mechanism of BMP gradient formation" for consideration by *eLife*. Your article has been reviewed by three peer reviewers, and the evaluation has been overseen by a Reviewing Editor and Robb Krumlauf as the Senior Editor. The reviewers have opted to remain anonymous.

The reviewers have discussed the reviews with one another and the Reviewing Editor has drafted this decision to help you prepare a revised submission.

The authors use quantitative imaging and modeling to come to the conclusion that a source-sink mechanism underlies the BMP/Chordin patterning system. This is novel as formation of a BMP gradient by a source-sink mechanism is distinct from previously reported models such as transcriptional, shuttling, or counter-gradient mechanisms. The measurement of BMP2 diffusion in vivo is novel. The topic is not only of broad interest but also quite controversial, and this study brings some much needed clarity to the field. The main weakness is the limited attempt to minimize the parameter space used for the modeling. Some of the measurements are difficult but there are several key measurements that need to be made. The computational and experimental analyses are incomplete, and some details of simulations have not been provided. For their model to be made more convincing a variety of concerns/comments should be fully addressed.

Major specific points:

1) The source-sink model is puzzling in light of previous reports showing that zebrafish and *Xenopus* chordin mRNA injections at the one-cell stage can fully rescue zebrafish chordino mutants (Fisher and Halpern, 1999; Schulte-Merker et al., 1997). In these embryos, injected chordin mRNA is present throughout the embryo and yet, the rescued mutants develop normally to adulthood (and presumably form a normal BMP gradient). The mRNA rescue experiments also call into question the notion that the transcriptional pattern is critical for BMP gradient formation. The authors should discuss the various models in light of these reports.

2) Relevant to the above, did the authors measure the effective diffusion of BMP2 by FRAP in chordin mutant embryos? This will provide experimental evidence to test the models.

3) Cell numbers increase substantially between 4 and 7 hpf, when the authors report a steep increase in the BMP gradient. How does the change in cell number in the zebrafish gastrula (between 4-7 hpf) compare to that during fly DV patterning? To what extent does this account for the increase in gradient slope in zebrafish?

4) The authors state "The equations were simulated 1,000,000 times".

Table 1 lists 15 free parameters; therefore each parameter can, in principle, only take ~2.5 different values when combined randomly (2.52^15 is ~10^6). However, the plots in Figure 4 suggest that many more values have been chosen for individual parameters. The authors should clarify how the parameters were varied in the simulations.

5) The authors state "To our surprise, there are greater than 50 times the number of source-sink to counter-gradient modeling solutions (Figure 6'), suggesting that the source-sink mechanism predominates." However, the number of different parameter combinations that can explain the data cannot be taken as a measure of support for a specific modelling solution. Different parameter combinations can result in similar solutions simply because the model is not identifiable or because it is difficult to find a unique solution when trying to fit the data directly.

6) If the authors already have or are making a BMP-destabilized GFP/degradFP fusion, precise measurement of BMP2 decay rates would be useful to distinguish between the models. If these are available it would be useful but not essential for the paper.

7) Why were the equations solved for the developmental window from 3.5 to 5.7 hpf and not from 4.7 to 6.7 hpf, when the pSMAD intensities were measured?

8) The authors presumably solved the equations in 1D in order to sample parameter space in a reasonable amount of time. I think it would add to the paper if the authors showed that the same parameters found to satisfy the 1D case would also set up the 3D P-Smad gradient when solving the equations in the hemispherical geometry of the zebrafish embryo as in Zhang, Lander, and Nie, J. Theor. Biol., 2007.

9) Can the P-Smad5 gradient of the chordin heterozygotes be included? Why are they "not shown"? Does the model fit the chordin heterozygote P-Smad5 gradient?

10) The pSmad5 measurements are impressive but the models ultimately rest on the BMP transcript and (active and inactive) protein expression profiles. I understand that there are no good antibodies to detect BMP but minimally, the authors need to use now standard quantitative fluorescent in situ hybridization approaches to measure BMP transcript distribution.

11) The BMP FRAP experiment goes a long way to reduce parameter space but why not also measure BMP diffusion in the presence of Chordin? This experiment seems trivial but would greatly help to test the shuttling model.

12) The characterization of the bmp2-venus chimera is too superficial. Does it have the same range and activity as wild-type bmp2?

[Editors' note: further revisions were requested prior to acceptance, as described below.]

Thank you for resubmitting your work entitled "Systems biology derived source-sink mechanism of BMP gradient formation" for further consideration at *eLife*. Your revised article has been favorably evaluated by Marianne Bronner as Senior editor, a Reviewing editor, and three reviewers. =

The manuscript has been improved but there are some remaining issues that need to be addressed before acceptance, as outlined below. In the absence of measurements of BMP diffusion in the presence of chordin or Chordin protein expression, the alternate model Zinski et al. suggest for BMP gradient formation cannot be substantiated. To support their clams, it is crucial that the authors:i) determine how the BMP gradient forms in the presence of uniform chordin ii) show expression of tagged BMP / Chordin proteins at early gastrula stages in mutants injected with RNA. iii) clarify how random sampling was done.

Key comments:

1) In response to how BMP diffuses in the absence and presence of chordin (previous review comments 2 and 11), in the revised Zinski m/s, the authors examined BMP diffusion without chordin. However, they did not examine BMP diffusion with uniform chordin. This is a crucial test of the model which is missing.

2) In their explanation of how their data and preferred model can explain the previous reports of rescue of chordino mutants with uniform chordin injections, Zinski et al. contend that chordin RNA injections likely do not generate uniform Chordin protein expression in embryos. But no evidence is provided to support this view – either experimentally or in their simulations. (The triple morphant is not relevant to the comment).

3) Regarding how parameters were varied in their simulations, the authors state that they prefer random sampling as opposed to a regular grid.

Did the authors refine sampling depending on the outcomes? If solutions did not change in a region in parameter space, did they increase the sampling density?

---

## [Author Response]

Major specific points:

1) The source-sink model is puzzling in light of previous reports showing that zebrafish and Xenopus chordin mRNA injections at the one-cell stage can fully rescue zebrafish chordino mutants (Fisher and Halpern, 1999; Schulte-Merker et al., 1997). In these embryos, injected chordin mRNA is present throughout the embryo and yet, the rescued mutants develop normally to adulthood (and presumably form a normal BMP gradient). The mRNA rescue experiments also call into question the notion that the transcriptional pattern is critical for BMP gradient formation. The authors should discuss the various models in light of these reports.

The ubiquitous expression of *chordin* RNA, which can rescue *chordin* mutant embryos, likely does not generate a ubiquitous distribution of Chordin protein due to the presence of Tolloid and Bmp1a, two ventrally expressed metalloproteases that degrade Chordin protein. The rescue of embryos lacking Chordin, Tolloid, and Bmp1a using *chordin* RNA injection at the one-cell stage has not been attempted as it is a difficult experiment to execute, but we hypothesize that it would not be effective. We agree that the *chordin* expression domain is likely not critical to BMP gradient formation.

2) Relevant to the above, did the authors measure the effective diffusion of BMP2 by FRAP in chordin mutant embryos? This will provide experimental evidence to test the models.

We have now repeated our FRAP experiments in Chordin deficient embryos and have added it to the main text of the paper and Figure 8. The effective diffusivity of Bmp2b-Venus in *chd* deficient embryos matched the effective diffusivity we had measured in WT embryos. This new experiment validates our previous assumption that the majority of BMP-Venus observed in our WT FRAP experiments was not bound to Chordin, as expected due to the ventralized embryonic phenotype at 24 hpf (Figure 8 row 2). However, performing the FRAP experiment in *chd* deficient embryos assures that 100% of Bmp2b-Venus observed is not bound to Chordin, removing the potential impact of a subpopulation of Chordin-bound Bmp2b-Venus on the FRAP measurements in WT embryos.

3) Cell numbers increase substantially between 4 and 7 hpf, when the authors report a steep increase in the BMP gradient. How does the change in cell number in the zebrafish gastrula (between 4-7 hpf) compare to that during fly DV patterning? To what extent does this account for the increase in gradient slope in zebrafish?

Fly DV patterning takes place during stage 5. Mitosis is paused during stage 5 until gastrulation begins at stage 6. Cells do not divide at this time, but nuclear localization of P-Mad increases from nearly none to a steep gradient in less than 1 hour from early stage 5 to early stage 6 when gastrulation begins. Thus, in this context while there are no changes in nuclei density, the gradient of PMad starts broad and refines to a higher amplitude and sharper profile [1]. However, in the context of patterning the embryo termini in *Drosophila* via Torso RTK signaling, the nuclei are present in a syncytium in the absence of cell membranes, where the nuclear density plays a role in shaping the gradient of phosphorylated ERK at the termini [2]. The nuclei density doubles with each successive division providing an increase in nuclear trapping of dpERK at the most terminal positions, resulting in a steeper gradient with higher peak amplitude and reduced gradient extent or length over time. The zebrafish embryo is not a synctium, so this same nuclear trapping mechanism could not occur. Moreover, the gradient DV length is constant over this time period, also inconsistent with this type of mechanism (Figure 2).

From our cell counts in embryos from a single timecourse (Figure 2), we observe an approximately 70% increase in cell number from 4.7 to 6.7 hours post fertilization (hpf) and a 100% increase in nuclear P-Smad5 amount. Cell nuclei do not change significantly in size during this time [3]. The increase in cell number occurs throughout the embryo and is not restricted to a particular DV region. So half of the 70% increase in cell number occurs in lateral and dorsal regions, where the slope does not change over this time period (Figure 2). The increase in slope observed during the 2 hour period occurs over the ventral half of the embryo, which accounts for 35% of the increase in cell number, while the gradient peak doubles in amplitude. Thus an increase in cell number, via an unknown mechanism, could not account for the increase in slope. Additional support that cell number has little effect on gradient shape, we observed that the absolute number of cells at a given time point can vary by as much as 20% between different embryos within the same cross or between crosses (Figure 2) with no detectable change in gradient shape or phenotype.

4) The authors state "The equations were simulated 1,000,000 times".

Table 1 lists 15 free parameters; therefore each parameter can, in principle, only take ~2.5 different values when combined randomly (2.52^15 is ~10^6). However, the plots in Figure 4 suggest that many more values have been chosen for individual parameters. The authors should clarify how the parameters were varied in the simulations.

For each model iteration, parameters were selected from a uniform distribution in log space that covered four orders of magnitude within the physiological range for each parameter. Each parameter was selected independently of the remaining parameters. We prefer the coverage offered by this method, as opposed to a regular grid wherein each parameter has a set of discrete values and coverage at those values is complete. Random sampling is not susceptible to the “curse of dimensionality” that is common to regular grid spacing methods allowing for more efficient estimation of the NRMSD landscape in Figure 7), and random points fill in the parameter space more evenly than regular grid spacing that forces evaluations of the model within the same parameter hyperplane [4]. Random sampling of the biophysical parameters is common to these types of problems [5, 6].

5) The authors state "To our surprise, there are greater than 50 times the number of source-sink to counter-gradient modeling solutions (Figure 6'), suggesting that the source-sink mechanism predominates." However, the number of different parameter combinations that can explain the data cannot be taken as a measure of support for a specific modelling solution. Different parameter combinations can result in similar solutions simply because the model is not identifiable or because it is difficult to find a unique solution when trying to fit the data directly.

We agree with this point and have changed the sentence so it now reads: "To our surprise, the source-sink modeling solutions emerged more frequently within our computational screen than the counter-gradient solutions (Figure 7'). "

6) If the authors already have or are making a BMP-destabilized GFP/degradFP fusion, precise measurement of BMP2 decay rates would be useful to distinguish between the models. If these are available it would be useful but not essential for the paper.

We agree it would be useful, but these constructs are not currently available.

7) Why were the equations solved for the developmental window from 3.5 to 5.7 hpf and not from 4.7 to 6.7 hpf, when the pSMAD intensities were measured?

While dorsal-ventral cell fate for the head and trunk is largely specified from 4 to 7 hours post fertilization [7-10], *bmp* and *chordin* are first expressed after the mid-blastula transition (MBT) at 3 hpf [11-14]. We started our simulation at 3.5 hpf to account for the time needed for these proteins to be translated, folded, and secreted. We have now added that rationale to the manuscript.

8) The authors presumably solved the equations in 1D in order to sample parameter space in a reasonable amount of time. I think it would add to the paper if the authors showed that the same parameters found to satisfy the 1D case would also set up the 3D P-Smad gradient when solving the equations in the hemispherical geometry of the zebrafish embryo as in Zhang, Lander, and Nie, J. Theor. Biol., 2007.

We agree that this would add to the paper and the development of data-driven three dimensional models continues to be an effort of our work. We have made progress in developing data-driven 3D models and this is being worked on but not achievable within the timeframe of the revision. The primary scope of the modeling effort herein focused on broad coverage of potential mechanisms to provide an unbiased view of data-consistent mechanisms. At this time, the screen and scope of our study is only achievable with models with one space dimension. Moreover, the inputs to the system based on gene expression are distributed symmetrically across the embryo, so a 1D model should largely reflect one in 3D. We added the rationale for performing a 1D model now to the manuscript.

9) Can the P-Smad5 gradient of the chordin heterozygotes be included? Why are they "not shown"? Does the model fit the chordin heterozygote P-Smad5 gradient?

Thank you for the suggestion. We have now added a comparison of WT to *chordin* heterozygotes to the manuscript (Figure 6). There is no significant change in the P-Smad5 gradient or embryonic phenotype between WT and *chordin* heterozygous embryos (Figure 6). We also now required mathematical model solutions to fit our *chordin* heterozygous data, when the Chordin production rate was set to 50%. Forcing the model to fit our *chordin* heterozygous data decreased the overall number of solutions, but did not eliminate all solutions of either the source-sink or counter-gradient mechanism. The mathematical model in Figure 7–Figure 8 and associated text have been updated to include *chordin* heterozygous data.

10) The pSmad5 measurements are impressive but the models ultimately rest on the BMP transcript and (active and inactive) protein expression profiles. I understand that there are no good antibodies to detect BMP but minimally, the authors need to use now standard quantitative fluorescent in situ hybridization approaches to measure BMP transcript distribution.

*bmp2b* expression is now quantified by RNAScope fluorescent *in situ* hybridization and included in Figure 3. We have adjusted our BMP production gradient in our mathematical model to fit the measured *bmp2b* expression gradient and simulated the mathematical model an additional 1,000,000 times. The adjusted BMP production gradient increased the overall number of solutions, but did not modify our conclusions about the mechanisms. All modeling figures and data displays (Figure 4,Figure 5,Figure 7,Figure 9) have been updated incorporating the new data.

11) The BMP FRAP experiment goes a long way to reduce parameter space but why not also measure BMP diffusion in the presence of Chordin? This experiment seems trivial but would greatly help to test the shuttling model.

While we agree that measuring the diffusivity of BMP bound to Chordin would be very useful for further testing the shuttling model, the experiment is not feasible without extensive overexpression or the development of additional tools. To see Bmp2b-Venus, we must moderately overexpress it. To ensure all Bmp2b-Venus is bound to Chordin, we would need to heavily overexpress Chordin. Chordin has been shown to interact with Tsg [15], Bmper [16-18], Ont1 [19], and HSPGs [20]. By overexpressing both BMP and Chordin, these interacting factors, which may limit BMP-Chordin diffusion could be diluted out, causing FRAP to overestimate the diffusivity of the BMP-Chd complex. Since we do not have time to develop more sophisticated tools to test this, we think it is best not to include a double overexpression experiment.

12) The characterization of the bmp2-venus chimera is too superficial. Does it have the same range and activity as wild-type bmp2?

To assess whether Bmp2b-Venus is properly processed and remains attached to the Venus tag, we performed a Western Blot for Venus on embryos injected with *venus* or *bm2b-venus* RNA. We have added this experiment to the paper (Figure 8).

To further assess the activity and range of the Bmp2b-Venus chimera, we rescued embryos lacking Bmp2b with *bmp2b-venus* RNA. It has been previously reported that embryos lacking Bmp2b can be rescued using *bmp2b* RNA [21]. We were able to rescue embryos to a WT phenotype by injecting *bmp2b-venus* RNA (Figure 8, rows 5-7). This result has been added to the text (Figure 8). However, a direct comparison of *bmp2b* RNA activity and *bmp2b-Venus* activity is not possible, as RNA activity can vary from synthesis reaction to synthesis reaction. In addition to this we have now measured secreted Venus diffusion rates with our approach and added additional measurements for Bmp2b-Venus diffusion in *chordin* MO-injected embryos (Figure 8).

At this time we are unable to compare the range of Bmp2b to that of Bmp2b-Venus. Range has two components: diffusivity and decay rate. Measuring diffusivity via FRAP requires the protein in question to be tagged fluorescently. While decay rates of Bmp2b and Bmp2b-Venus could be measured and compared by injecting Bmp2b and Bmp2b-Venus protein and performing a western blot time-series, previous estimates of Tgf-β decay rates have relied exclusively on tagged proteins [22].

[Editors' note: further revisions were requested prior to acceptance, as described below.]

Key comments:

1) In response to how BMP diffuses in the absence and presence of chordin (previous review comments 2 and 11), in the revised Zinski m/s, the authors examined BMP diffusion without chordin. However, they did not examine BMP diffusion with uniform chordin. This is a crucial test of the model which is missing.

We presume that by requesting that we measure BMP diffusion with uniform chordin that the reviewers are asking us to measure bound BMP-Chd diffusion. We do assert that the Shuttling mechanism requires high diffusivity of BMP-Chordin (Figure 5 x-axis), however, our *chordin*-/- mutant data already exclude the Shuttling mechanism (Figure 6). However, when discussing the Source-Sink versus the Counter-Gradient mechanism in Figure 7, we omitted an important plot of BMP-Chd diffusivity in our latest submission, which addresses the reviewers’ point. We thank the reviewers for making us aware of this. A plot of BMP-Chd diffusivity shows that possible values of BMP-Chd diffusivity over the 4 orders of magnitude tested in our simulations does not distinguish between the Counter-Gradient and Source-Sink mechanisms. Both of these mechanisms are compatible with BMP-Chd diffusivities over 4 orders of magnitude from nearly immobile (10^-2^ µm^2^/s) to free diffusion (10^2^ µm^2^/s). Therefore, measuring BMP-Chd diffusivity would not help discern between the two remaining models, Counter-Gradient and Source-Sink. We added this BMP-Chd diffusivity plot now to the manuscript to clarify this important point (Figure 7).

2) In their explanation of how their data and preferred model can explain the previous reports of rescue of chordino mutants with uniform chordin injections, Zinski et al. contend that chordin RNA injections likely do not generate uniform Chordin protein expression in embryos. But no evidence is provided to support this view – either experimentally or in their simulations. (The triple morphant is not relevant to the comment).

In comment 1 of the previous review, the reviewers asked us to discuss the rescue of *chordino* mutants with uniform *chordin* RNA injections. We address this point now by simulating the system with ubiquitous Chordin production. We found that 426 of the 1452 solutions that fit our WT, *chordin*-/-, and *chordin*+/- data and BMP diffusivity (within 2 um^2^/s of our measured 4.4 um^2^/s) retain a WT BMP gradient when Chordinis uniformly produced. The 426 remaining solutions are all source-sink mechanisms with steep gradients of Chordin that are high dorsally and low ventrally. The results of the simulations show that the BMP gradient is generated by Tolloid degrading Chordin ventrally. We include these simulation results that show how uniform *chordin* RNA can rescue a *chordin* mutant in the Results section of our paper, as a new panel Figure 8. We thank the reviewers for the suggestion, as it provides further support for the source-sink mechanism.

3) Regarding how parameters were varied in their simulations, the authors state that they prefer random sampling as opposed to a regular grid.

Did the authors refine sampling depending on the outcomes? If solutions did not change in a region in parameter space, did they increase the sampling density?

For each model iteration, parameters were selected from a uniform distribution in log space that covered four orders of magnitude within the physiological range for each parameter. Each parameter was selected independently of the remaining parameters. Adaptive and subsampling methods that increase parameter selection in regions with high variance in model output were not used in our parameter selection for a number of reasons. Firstly, a parameter matrix is produced that is then subdivided across distributed computers to solve the pdes to produce a stored file of model solutions and parameter vectors associated with the stored solution. For each parameter vector, the PDE system is converted into a set of 180 ordinary differential equations after discretization and dynamically solved for wt, chd-/-, nog-/-, and chd+/- conditions using the implicit solver ode15s. Ode15s is well suited to problems with numerical stiffness that arise during numerical screens with random parameters. Thus, for an ensemble of 1 million parameter vectors solved, the system of differential equations are solved 4 times to simulate the wt and mutant conditions, increasing the total model evaluations to 4 million. Following calculation of the model solutions for each parameter vector, post processing, sorting, and calculation of model fitness against the data is handled by a separate program that operates on the stored solutions. The separation of model evaluation from model analysis allows for much greater flexibility, the total number of simulation results, and an ability to add additional simulation results to existing simulations that have previously been computed. If a job is interrupted, the index and solutions up to the parameter index are stored and simulation jobs can be restarted at the last iteration point.

Random sampling is not susceptible to the “curse of dimensionality” that is common to regular grid spacing methods allowing for more efficient estimation of the NRMSD landscape in Figure 7), and random points fill in the parameter space more evenly than regular grid spacing that forces evaluations of the model within the same parameter hyperplane (Caflisch et al., 1998). Figure 7 demonstrates the dense coverage and sampling to estimate the NRMSD values and dense coverage in regions with acceptable NRMSD values. Random sampling of the biophysical parameters is common to these types of problems (Eldar et al., 2002; von Dassow et al., 2000).

We have added this explanation to the Materials and methods section as further clarification.

1. Wang, Y. and E.L. Ferguson, *Spatial bistability of Dpp–receptor interactions during Drosophila dorsal–ventral patterning.* Nature, 2005. 434(7030): p. 225-9.

2. Coppey, M., et al., *Nuclear trapping shapes the terminal gradient in the Drosophila embryo.* Curr Biol, 2008. 18(12): p. 915-9.

3. Keller, P.J., et al., *Reconstruction of zebrafish early embryonic development by scanned light sheet microscopy.* Science, 2008. 322(5904): p. 1065-9.

4. Caflisch, A., R. Walchli, and C. Ehrhardt, *Computer-Aided Design of Thrombin Inhibitors.* News Physiol Sci, 1998. 13: p. 182-189.

5. Eldar, A., et al., *Robustness of the BMP morphogen gradient in Drosophila embryonic patterning.* Nature, 2002. 419(6904): p. 300-4.

6. von Dassow, G., et al., *The segment polarity network is a robust developmental module.* Nature, 2000. 406.

7. Tucker, J.A., K.A. Mintzer, and M.C. Mullins, *The BMP signaling gradient patterns dorsoventral tissues in a temporally progressive manner along the anteroposterior axis.* Dev Cell, 2008. 14(1): p. 108-19.

8. Hashiguchi, M. and M.C. Mullins, *Anteroposterior and dorsoventral patterning are coordinated by an identical patterning clock.* Development, 2013. 140(9): p. 1970-80.

9. Tuazon, F.B. and M.C. Mullins, *Temporally coordinated signals progressively pattern the anteroposterior and dorsoventral body axes.* Semin Cell Dev Biol, 2015. 42: p. 118-33.

10. Kwon, H.J., et al., *Identification of early requirements for preplacodal ectoderm and sensory organ development.* PLoS Genet, 2010. 6(9): p. e1001133.

11. Leung, T., *bozozok directly represses bmp2b transcription and mediates the earliest dorsoventral asymmetry of bmp2b expression in zebrafish.* Development, 2003. 130(16): p. 3639-3649.

12. Solnica-Krezel, L. and W. Driever, *The role of the homeodomain protein Bozozok in zebrafish axis formation.* Int J Dev Biol, 2001. 45(1): p. 299-310.

13. Koos, D. and R. Ho, *The nieuwkoid dharma homeobox gene is essential for bmp2b repression in the zebrafish pregastrula.* Developmental Biology, 1999. 215(2): p. 190–207.

14. Shimizu, T., et al., *Cooperative roles of Bozozok/Dhama and Nodal-Related protein in the formation of the dorsal organizer in zebrafish.* Mech Dev, 2000. 91(1-2): p. 293-303.

15. Troilo, H., et al., *Structural characterization of twisted gastrulation provides insights into opposing functions on the BMP signalling pathway.* Matrix Biol, 2016. 55: p. 49-62.

16. Rentzsch, F., et al., *Crossveinless 2 is an essential positive feedback regulator of Bmp signaling during zebrafish gastrulation.* Development, 2006. 133(5): p. 801-11.

17. Zhang, J.L., et al., *Binding between Crossveinless-2 and Chordin von Willebrand factor type C domains promotes BMP signaling by blocking Chordin activity.* PLoS One, 2010. 5(9): p. e12846.

18. Ambrosio, A.L., et al., *Crossveinless-2 Is a BMP feedback inhibitor that binds Chordin/BMP to regulate Xenopus embryonic patterning.* Dev Cell, 2008. 15(2): p. 248-60.

19. Inomata, H., T. Haraguchi, and Y. Sasai, *Robust stability of the embryonic axial pattern requires a secreted scaffold for chordin degradation.* Cell, 2008. 134(5): p. 854-65.

20. Jasuja, R., et al., *Cell-surface heparan sulfate proteoglycans potentiate chordin antagonism of bone morphogenetic protein signaling and are necessary for cellular uptake of chordin.* J Biol Chem, 2004. 279(49): p. 51289-97.

21. Nguyen, V.H., et al., *Ventral and lateral regions of the zebrafish gastrula, including the neural crest progenitors, are established by a bmp2b/swirl pathway of genes.* Dev Biol, 1998. 199(1): p. 93-110.

22. Muller, P., et al., *Differential diffusivity of Nodal and Lefty underlies a reaction-diffusion patterning system.* Science, 2012. 336(6082): p. 721-4.